# Thermal non-equilibrium of porous flow in a resting matrix applicable to melt migration: a parametric study

Laure Chevalier, Harro Schmeling

Institute of Geosciences, Goethe University, 60438 Frankfurt, Germany

*Correspondence to*: Harro Schmeling (schmeling@geophysik.uni-frankfurt.de)

**Abstract.** Fluid flow through rock occurs in many geological settings on different scales, at different temperature conditions and with different flow velocities. Depending on these conditions the fluid will be in local thermal equilibrium with the host rock or not. To explore the physical parameters controlling thermal non-equilibrium the coupled heat equations for fluid and solid phases are formulated for a fluid migrating through a resting porous solid by porous flow. By non-dimensionalizing the

equations two non-dimensional numbers can be identified controlling thermal non-equilibrium: the Peclet number $Pe$ describing the fluid velocity, and the porosity $\phi$. The equations are solved numerically for the fluid and solid temperature evolution for a simple 1D model setup with constant flow velocity. This setup defines a third non-dimensional number, the initial thermal gradient $G$, which is the reciprocal of the non-dimensional model height $H$. Three stages are observed: a transient stage followed by a stage with maximum non-equilibrium fluid to solid temperature difference, $\Delta T_{max}$, and a stage

approaching the steady state. A simplified time-independent ordinary differential equation for depth-dependent $(T_f - T_s)$ is derived and solved analytically. From these solutions simple scaling laws of the form $(T_f - T_s) = f(Pe, G, z)$ are derived. Due to scaling they don't depend explicitly on $\phi$ anymore. The solutions for $\Delta T_{max}$ and the scaling laws are in good agreement with the numerical solutions. The parameter space $Pe, G$ is systematically explored. Three regimes can be identified: 1) at high $Pe$ (>1/G) strong thermal non-equilibrium develops independently of $Pe$; 2) at low $Pe$ (<1/G) non-equilibrium decreases

proportional to decreasing $Pe \cdot G$; 3) at low $Pe$ (<1) and $G$ of order 1 the scaling law is $\Delta T_{max} \approx Pe$. The scaling laws are also given in dimensional form. The dimensional $\Delta T_{max}$ depends on the initial temperature gradient, the flow velocity, the melt fraction, the interfacial boundary layer thickness, and the interfacial area density. The time scales for reaching thermal non-equilibrium scale with the advective time-scale in the high $Pe$-regime and with the interfacial diffusion time in the other two low $Pe$ - regimes. Applying the results to natural magmatic systems such as mid-ocean ridges can be done by estimating

appropriate orders of $Pe$ and $G$. Plotting such typical ranges in the $Pe$ - $G$ regime diagram reveals that a) interstitial melt flow is in thermal equilibrium, b) melt channeling such as e.g. revealed by dunite channels may reach moderate thermal non-equilibrium with fluid to solid temperature differences of up to several 10's Kelvin, and c) the dyke regime is at full thermal non-equilibrium.

# 1 Introduction

Fluid flow through rock occurs in many geological settings on different scales, at different temperature conditions and with different flow velocities. Depending on these conditions the fluid will be in local thermal equilibrium with the host rock or not. On a small scale, e.g. grain scale, usually thermal equilibrium is valid. Examples include melt migration through a porous matrix in the asthenosphere or in crustal magmatic systems at super-solidus temperatures (e.g. McKenzie, 1984), groundwater or geothermal flows in sediments or cracked rocks (e.g. Verruijt, 1982; Furbish, 1997; Woods, 2015), or hydrothermal

convection in the oceanic crust (e.g. Davis et al., 1999; Harris and Chapman, 2004; Becker and Davies, 2004). On a somewhat larger scale local thermal equilibrium may not be reached always. Examples of such flows include melt migration in the mantle or crust at temperatures close to or slightly below the solidus where melt may be focused and migrates through systems of veins or channels (Kelemen et al., 1995; Spiegelman et al., 2001). Within the upper oceanic crust also water may migrate through systems of vents or channels (Wilcock and Fisher, 2004). At even larger scales and at sub-solidus conditions magma

rapidly flows through propagating dykes or volcanic conduits (e.g. Lister and Kerr, 1991; Rubin, 1995; Rivalta et al., 2015) and is locally at non-equilibrium with the host rock.

Heat transport associated with most of such flow scenarios is usually described by assuming thermal equilibrium between the fluid and solid under slow flow conditions (e.g. McKenzie 1984). Alternatively, for more rapid flows such melts moving in dykes through a cold elastic or visco-elasto-plastic ambient rock, the fluids are assumed as isothermal (e.g. Maccaferri et al.,

2011; Keller et al., 2013). However, on local scale of channel or dyke width thermal interaction between rising hot magma and cold host rock is important and may lead to effects such as melting of the host rock and freezing of the magma with important consequences for dyke propagation and the maximum ascent height (e.g. Bruce and Huppert, 1990; Lister and Kerr, 1991; Rubin, 1995). Clearly, in such rapid fluid flow scenarios melt is not in thermal equilibrium with the ambient rock.

Thus, there exists a transitional regime, which, for example, may be associated with melt focusing into pathways where flow

is faster and thermal equilibrium might not be valid anymore. In such a scenario it might be possible that channelized flow of melt might penetrate deeply into sub-solidus ambient rock, and thermal non-equilibrium delays freezing of the ascending melts and promotes initiation of further dyke-like pathways. Indeed, for mid-oceanic ridges compositional non-equilibrium has proven to be of great importance for understanding melt migration and transport evolution (Aharonov et al., 1995; Spiegelman et al., 2001). Thus, it appears plausible that in cases of sufficiently rapid fluid flow e.g. due to channeling or fracturing thermal

non-equilibrium may also become important. Describing this non-equilibrium macroscopically, i.e. on a scale larger than the pores or channels, is the scope of this paper.

While the physics of thermal non-equilibrium in porous flow is well studied in more technical literature (e.g. Schumann, 1929; Spiga and Spiga, 1981; Kuznetsov, 1994; Amiri and Vafai, 1994; Minkowycz et al., 1999; Nield and Bejan, 2006; de Lemos, 2016), so far it has attracted only little attention in the geoscience literature, but see Schmeling et al., (2018) and Roy (2020).

The basic approach in all these studies is the decomposition of the heat equation for porous flow into two equations, one for the solid and one for the migrating fluid. The key parameter for thermal non-equilibrium is a heat exchange term between fluid

and solid, which appears as a sink in the equation for the fluid and as a source in the equation for the solid. Usually, this heat exchange term is assumed proportional to the local temperature difference between fluid and solid (Minkowycz et al. 1999; Amiri and Vafai, 1994; de Lemos, 2016; Roy, 2020). However, Schmeling et al. (2018) showed that in a more general formulation the heat exchange term depends on the complete thermal history of the moving fluid through the possibly also moving solid. Here we will follow the common assumption and use the local temperature difference formulation. While Schmeling et al. (2018) showed that the magnitude of thermal non-equilibrium essentially depends on the flow velocity, or more precisely, on the Peclet number, here we will more generally explore the parameter space.

While thermal non-equilibrium of an arbitrary porous flow system depends on many parameters, our approach is to reduce the complexity of the system and systematically explore the non-dimensional parameter space. It will be shown that only two non-dimensional parameters control thermal non-equilibrium in porous flow, namely the Peclet number and the porosity. In our simple 1D model setup with constant flow velocity a third non-dimensional number, the non-dimensional initial thermal gradient $G$, is identified, which is equal to the reciprocal non-dimensional model height $H=1/G$. The non-dimensionalization allows application of the results to arbitrary magmatic or other systems. The aim is to derive scaling laws that allow an easy determination of whether thermal equilibrium or non-equilibrium is to be expected and quantitatively to estimate the maximum temperature difference between fluid and matrix. The results will be applied the magmatic system of a mid-ocean ridge setting.

## 2 Governing equations and model setup

### 2.1 Heat conservation equations

We start with considering a general two-phase matrix-fluid system with variable properties and solid and fluid velocities and subsequently apply simplifications. The two phases are incompressible, and we assume local thermal non-equilibrium conditions, i.e. the two phases exchange heat. The equations for conservation of energy of this system are given e.g. by de Lemos (2016). Assuming constant pressure the conservation of energy of the fluid phase is given by:

$$c_{p,f}\left(\frac{\partial(\phi\rho_f T_f)}{\partial t} + \nabla \cdot \left(\phi\rho_f v_f T_f\right)\right) = \nabla \cdot \left(\phi\lambda_f \nabla T_f\right) - Q_{fs} \tag{1}$$

For the definition of all quantities, see Table 1. Equation (1) can be rearranged into:

$$c_{p,f}\left(T_f \frac{\partial(\phi\rho_f)}{\partial t} + \phi\rho_f \frac{\partial T_f}{\partial t} + T_f \nabla \cdot \left(\phi\rho_f v_f\right) + \phi\rho_f v_f \cdot \nabla T_f\right) = \nabla \cdot \left(\phi\lambda_f \nabla T_f\right) - Q_{fs} \tag{2}$$

Mass conservation for the fluid phase is given by:

$$\frac{\partial(\rho_f \phi)}{\partial t} + \nabla \cdot \left(\rho_f \phi v_f\right) = 0 \tag{3}$$

Inserting (3) into (2), conservation of energy for the fluid phase becomes:

$$c_{p,f}\rho_f \phi \left(\frac{\partial T_f}{\partial t} + v_f \cdot \nabla T_f\right) = \nabla \cdot \left(\phi\lambda_f \nabla T_f\right) - Q_{fs} \tag{4}$$

In a similar way, the conservation of energy of the solid phase is given by:

$$c_{p,s}\rho_s(1-\phi)\left(\frac{\partial T_s}{\partial t} + v_s \cdot \nabla T_s\right) = \nabla \cdot \left((1-\phi)\lambda_s \nabla T_s\right) + Q_{fs} \tag{5}$$

which, assuming that $v_s = 0$, is further simplified:

$$c_{p,s}\rho_s(1 - \phi)\frac{\partial T_s}{\partial t} = \nabla \cdot \left((1 - \phi)\lambda_s\nabla T_s\right) + Q_{fs} \tag{6}$$

The term $Q_{fs}$ in the fluid and solid heat conservation equations is the interfacial heat exchange term between the two phases
(fluid and solid). In general, it depends on the local thermal history of the two phases and the history of the heat exchange
(Schmeling et al., 2018). In a simplification it can be written as a combination of the interfacial area density $S$ with the
dimension [1/m], the interfacial boundary layer thickness $\delta$, the effective thermal conductivity $\lambda_{eff}$ and the temperatures of the
two phases:

$$Q_{fs} = \frac{S\lambda_{eff}}{\delta}(T_f - T_s) \tag{7}$$

In general, the term $\delta$ is time dependent. Schmeling et al. (2018), however, provide evidence that taking an appropriate constant
value for $\delta$ (depending on fluid velocity) gives a good approximation of $Q_{fs}$ and allows for reasonable modeling of temperature
evolution with time. In most of the following parametric study, we use this simplification for $\delta$ by assuming it is constant with
time.

## 2.2 Scaling and non-dimensionalization

Non-dimensionalization is useful for interpreting models involving a large number of parameters. It usually helps reducing the
number of parameters, and identifies non-dimensional parameters that control the evolution of the system. We write the two
energy conservation equations in a non-dimensional form, using

$$T = \Delta T_0 T', \ t = t_0 t', \ v = v_{f0}v', \ (x,y,z) = L \cdot (x',y',z') \tag{8}$$

where $\Delta T_0$ is the macroscopic scaling temperature difference of the system, i.e. the initial temperature difference between top
and bottom, $x,y,z$ is a distance, $v_{f0}$ is the scaling fluid velocity, $L$ is the scaling length

$$L = \sqrt{\frac{\phi_0(1-\phi_0)\delta}{S}} \tag{9}$$

with $\phi_0$ as a scaling porosity, and $t_0$ is the scaling time based on the diffusion time over the length $L$,

$$t_0 = L^2/\kappa_0 \tag{10}$$

(see Table 1 for definitions). Primed quantities are non-dimensional.

Introducing the fluid filled pore width $d_f$ and the solid width $d_s$ which may be the grain size or distance between fluid channels,
the interfacial area density $S$ scales with

$$S = \frac{c\phi_0}{d_f} \tag{11}$$

for melt channels, tubes, pockets for all melt fractions, and for melt films at small melt fractions, while $S$ scales with

$$S = \frac{c_s(1-\phi_0)}{d_s} \tag{12}$$

for melt channels, films and suspensions at all melt fractions. Here $c$ is a geometrical constant of the order 2 for melt channels,
4 for melt tubes, 6 for melt pockets, and 2 for melt films at small melt fractions. It should be emphasized that eqs. (11) and

(12) are different ways of calculating the same $S$. The geometrical constant $c_s$ is of order 2 for melt channels, and 6 for melt films or suspensions. Thus, the scaling time and scaling length can also be written as

$$t_0 = \frac{(1-\phi_0)d_f\delta}{c\kappa_0} = \frac{\phi_0 d_s \delta}{c_s \kappa_0} \tag{10a}$$

and

$$L = \sqrt{\frac{(1-\phi_0)\delta d_f}{c}} = \sqrt{\frac{\phi_0 \delta d_s}{c_s}} \tag{9a}$$

Eq. (9a) shows that $L$ scales both with the geometric mean of $d_f$ and $\delta$ at small melt fractions, and with the geometric mean of $d_s$ and $\delta$ at large melt fractions. Thus, $L$ is a natural length scale associated with thermal equilibrium of fluid filled pores. The above scaling laws for $S$ justify using the term $\phi_0(1-\phi_0)$ in the scaling length $L$. It should be noted that we introduce

and understand $d_s$ as the average distance between melt filled pores or channels which can be considerably larger than the grain size. Then both, $\delta$ and $d_s$, and thus $L$ can reach some considerable fraction of the system dimension.

We assume that the fluid and solid phases have the same densities and thermal properties (but relax this assumption later in section 5.1.3):

$$c_{p,f} = c_{p,s} = c_{p,0}, \quad \rho_f = \rho_s = \rho_0, \quad \kappa_f = \kappa_s = \frac{\lambda_{eff}}{c_{p,0}\rho_0} = \kappa_0 \tag{13}$$

From Eq. (4), (6), and (7) we get the non-dimensional energy conservation equations for the fluid and solid phases, respectively:

$$\phi\left(\frac{\partial T_f'}{\partial t'} + Pe\, v_f' \cdot \nabla T_f'\right) = \nabla \cdot \left(\phi \nabla T_f'\right) - \phi_0(1-\phi_0)(T_f' - T_s') \tag{14}$$

$$(1-\phi)\frac{\partial T_s'}{\partial t'} = \nabla \cdot \left((1-\phi)\nabla T_s'\right) + \phi_0(1-\phi_0)(T_f' - T_s') \tag{15}$$

From these equations we notice that the thermal evolution of the two-phase system is controlled by two non-dimensional

numbers: the scaling porosity $\phi_0$ and the Peclet number $Pe$ defined as

$$Pe = \frac{v_{f0}L}{\kappa_0} \tag{16}$$

This number has already proven to be of high significance for determining whether thermal non-equilibrium is present or not (Schmeling et al. 2018), and the highest $Pe$ corresponds to the largest temperature difference between fluid and matrix. In the following we drop the primes keeping all equations non-dimensional, if not indicated otherwise.

In the following we consider a homogeneous two-phase matrix-fluid system in 1D with a porosity constant in space and time, i.e. $\phi = \phi_0$. We assume a constant fluid velocity which will be expressed in terms of $Pe$, thus we choose the non-dimensional velocity $v_f = 1$. This simplifies equations (14) and (15) to

$$\frac{\partial T_f}{\partial t} + Pe \frac{\partial T_f}{\partial z} = \frac{\partial^2 T_f}{\partial z^2} - (1-\phi_0)(T_f - T_s) \tag{17}$$

and

$$\frac{\partial T_s}{\partial t} = \frac{\partial^2 T_s}{\partial z^2} + \phi_0(T_f - T_s), \tag{18}$$

respectively. As we are interested in the evolution of the non-equilibrium temperature difference between the solid and fluid, subtraction of Eq. (18) from Eq. (17) gives:

$$\frac{\partial(T_f - T_s)}{\partial t} - \frac{\partial^2(T_f - T_s)}{\partial z^2} + Pe\frac{\partial T_f}{\partial z} + (T_f - T_s) = 0 \tag{19}$$

which is equivalent to:


$$\frac{\partial(T_f - T_s)}{\partial t} - \frac{\partial^2(T_f - T_s)}{\partial z^2} + Pe\frac{\partial(T_f - T_s)}{\partial z} + (T_f - T_s) = -Pe\frac{\partial T_s}{\partial z} \tag{20}$$

Note that while the temperatures $T_f$ and $T_s$ explicitly depend on two non-dimensional numbers $Pe$ and $\phi_0$, the temporal evolution of the temperature difference $(T_f - T_s)$ explicitly depends only on $Pe$. However, implicitly it is still a function of $\phi_0$ because $T_s$ on the right-hand-side of Eq. (20) depends on $\phi_0$ via Eq. (18). Only for cases or stages with $T_s$ independent of $\phi_0$ as proposed in section 4, the temperature difference $(T_f - T_s)$ is a function of only one non-dimensional parameter, $Pe$,

and no more of $\phi_0$.

## 2.3 Model setup

The fluid and solid heat conservation equations are solved in a 1D domain of height $H$. Other geometries could also be easily explored but are not considered here, since we focus on studying the relative control of the scaling parameters on thermal non-equilibrium evolution. At time $t < 0$, both solid and fluid are at rest, in equilibrium. Both initial temperatures decrease linearly

from 1 to 0 with $z$, therefore a constant temperature gradient of $-G = -1/H$ is present in both phases (see Fig. 1). As boundary condition both phases temperatures are set to 1 (non-dimensional temperature) at $z = 0$. At $z = H$ a constant thermal gradient condition $\partial T/\partial z = -1/H$ (non-dimensional) is imposed for both phases. At $z = 0$ the advective flux is fixed by the constant temperature condition, i.e. it is equal to $Pe\,\phi_0$, while at $z = H$ it evolves freely with the fluid temperature, i.e. it is given by $T_f Pe\,\phi_0$ (all non-dimensional). This top boundary condition needs some justification: The hyperbolic partial differential

equations Eq. (17) or (18) require two well defined boundary conditions each, Dirichlet (fixed temperature), Neumann (fixed thermal gradient), Robin (linear combination of Neumann and Dirichlet) or Cauchy (fixed temperature *and* thermal gradient). Applying the Dirichlet condition at the bottom, leaves either a Dirichlet, a Neumann or a Robin condition to specify for the top. Different combinations of these boundary conditions can be applied separately for the fluid and the solid. For example, for the fluid a Neumann condition with zero or small temperature gradient may be reasonable, while for the solid one may

consider a Robin boundary condition mimicking a thick conductive lid with an internal constant temperature gradient and a fixed surface temperature. However, temporal changes of the temperature at the top (of our system = bottom of the lid) would lead to unphysical variations of the constant slope of the temperature gradient within the imagined lid. This is because the temperature in the lid can only vary on the diffusive timescale of the lid, which is much longer than all timescales in our model as long as the lid is thicker than $H$. In fact, in an open outflow situation like our system neither the evolution of the temperature,

the thermal gradient nor the total (advective plus conductive) heat flux is known a priori, but depends on the evolution within the system. In the early stage of model evolution both the solid and fluid have a thermal gradient inherited from the initial

condition which is advected upwards in the fluid. Thus it seems most appropriate to use the Neumann condition as a boundary condition for both the solid and fluid. Only at later stages this boundary condition imposes artefacts in the temperatures field close to the top boundary. The limitations of this top boundary condition are tested and discussed in chapter 5.1.2.

This model setup adds a third non-dimensional scaling parameter to the system, namely $G = 1/H$. It defines the initial non-dimensional temperature gradient or conductive heat flux, positive for a flux directed upwards. To summarize, the temperatures depend on the non-dimensional parameters $Pe$, $\phi_0$, and $G$.

## 2.4 Numerical scheme

The equations are solved by a MATLAB (MATLAB R2021b) code using a finite difference scheme central in space for the
conduction terms, upwind for the advection term, and explicit in time. The spatial resolution is $dz = 0.1$ or, for a few cases in Fig. 3 below, $Min(0.1, H/100)$ for $H < 10$. The time step was chosen as $dt = \frac{1}{4}Min(dz/Pe, dz^2)$, i.e. taking the minimum of the Courant or diffusion criterion. Tests with higher spatial and temporal resolution have been carried out and did not change the results visibly. The global heat balance has also been checked: The maximum relative heat balance error can be defined as

$$\delta_q = \left. \frac{q_{tot}(z=0) - q_{tot}(z=H) - H\frac{\partial T_{mean}}{\partial t}}{(q_{tot}(z=0) + q_{tot}(z=H))/2} \right|_{max} \text{, where } q_{tot} \text{ is the total non-dimensional vertical heat flux (conductive and advective)}$$

and $T_{mean}$ is the mean temperature of the model. $\delta_q$ has an error order 1 (due to the upwind scheme) with respect to the grid size $dz$, i.e. it is approximately equal to $const \cdot dz$ where the constant is of the order 0.2 (i.e. 2%) for high Peclet numbers and drops to 0.1 (1%) or 0.01 (0.1%) for $Pe \cong 1$ or smaller, respectively.

## 3 Numerical model results

First, some example numerical results are shown in Fig. 2 to understand the physics and the typical behavior.

## 3.1 Evolution of temperatures and thermal non-equilibrium with time

Three different models have been run, all with $Pe = 1$ and the following other parameters: Model 1: $G = 0.1$ ($H = 10$), $\phi = 0.1$, model 2: $G = 0.01$ ($H = 100$), $\phi = 0.1$, and Model 3: $G = 0.01$ ($H = 100$), $\phi = 0.2$. Figure 2a and b show $T_f$ and $T_s$ as functions of $z$ at different times as indicated for two initial temperature gradients, $G = 0.1$ ($H = 10$) and $G = 0.01$ ($H = 100$), respectively. Figure 2c shows the different contributions to the depth-dependent conductive and advective heat fluxes through
the solid and fluid phase, respectively. Figure 2e shows the evolution of $T_f$ and $T_s$ with time at the top of the domain, for the same model 2 as in Figure 2b and for model 3 with a higher melt fraction $\phi = 0.2$. Figure 2f shows the evolution of ($T_f$ - $T_s$) at different distances $z$ of model 2 ($\phi = 0.1$) and of model 3 ($\phi = 0.2$). At each depth of the system, the fluid and solid temperatures, as well as the temperature difference and the heat fluxes, evolve following three stages:

**Stage 1:** During this transient stage the fluid temperature increases faster than the solid temperature (Fig. 2a,b,e,f), and the
temperature difference (Fig. 2f,h) increases. During this stage, the fluid temperature increases rapidly at first, then the

temperature increase slows down. The conductive heat fluxes in both solid and fluid decrease rapidly and more slowly later, while the advective heat flux rapidly increases. As for the solid temperature, it first increases slowly, then faster and faster. At $t = 0$, the fluid velocity is suddenly set to non-zero, thus the fluid temperature starts to deviate from equilibrium and increases due to these new conditions. If the solid temperature were maintained constant with time, the fluid temperature would probably reach a steady state profile, depending on boundary conditions, fluid velocity and solid temperature. While the fluid temperature increases faster than the solid temperature, the fluid-solid temperature difference, thus the heat transfer term, increases too, forcing the solid temperature to progressively increase. At the end of stage 1 the maximum temperature difference is approached (Fig. 2h). Because the solid temperature hasn't risen significantly at that time (at $t = 4$ in the example) compared to the fluid temperature (Fig. 2g) different melt fractions do not affect the temperature differences during this stage (Fig. 2h in which all curves merge in one curve). This observation confirms the expectation from Eq. (20) that the temperature difference does not depend on melt fraction as long as the solid temperature is independent of $\phi$, which is the case as long as $T_s$ stays close to its initial profile.

**Stage 2:** The fluid and the solid temperatures increase at similar rates, constant with time (Fig. 2e), the temperature difference remains constant and at maximum at the top (Fig. 2f). Solid-fluid heat transfer is at maximum during this stage. As $T_s$ is no longer constant in time, different melt fractions lead to different rates of temperature increase (Fig. 2e) and also to different evolutions of $(T_f - T_s)$ (Fig. 2f solid curves compared to dashed curves). At higher melt fraction the heat transfer into the solid increases (c.f. last term in Eq. 18), resulting in a faster increase of the solid temperature whose gradient flattens earlier. Thus, the end of stage 2 is reached earlier (Fig. 2e).

**Stage 3:** As the fluid temperature rises closer to the $T_f$ - value at the bottom, its increase slows down, and heat transfer, thus temperature difference, decreases. In model 1 (Fig. 2a), steady state is reached while the fluid and solid temperatures are still far from 1. This is due to the influence of boundary conditions, as the heat transferred from the fluid phase to the solid phase is compensated by the solid phase heat loss at the top of the domain. In model 2 (Fig. 2b), boundary conditions at $z = H$ are applied farther away from the bottom, therefore allowing for a higher increase of temperatures when reaching the steady state. At each $z$ we observe that the temperature difference first increases rapidly to reach a maximum after a short time (stage1), here after $t = 4$ (Fig. 2h). The resulting amplitude of the temperature difference is identical at the different z-positions and for both melt fractions. Then it stays constant at this maximum value (stage 2), and finally decreases (stage 3) (Fig. 2f). The higher in the model, the longer the temperature difference remains at maximum. A higher melt fraction accelerates the decrease of $(T_f - T_s)$.

The absolute maximum temperature difference in space and time does not depend on boundary conditions (see also section 5.1.2 where the influence of boundary conditions is discussed), nor on the z-position nor on the melt fraction and therefore looks to be an interesting observable. It could indeed be useful for getting a first order estimate of thermal non-equilibrium conditions and possible temperature differences in a magmatic system. In the following sections we study how this maximum temperature difference evolves when varying the parameter $Pe$.

Comparing the heat fluxes of model 1 ($G = 0.1$) with those of model 2 ($G = 0.01$) shows the importance of heat advection by the fluid phase: In model 1 (Fig. 2c) the conductive contribution through the solid is of same order of magnitude as the advective contribution by the fluid because the initial temperature gradient $G$ and the porosity $\varphi$ are the same (= 0.1). In model 2 (Fig. 2d), in spite of the same Peclet number, the smaller initial temperature gradient ($G = 0.01$) reduced the conductive with respect to the advective contribution by a factor of about 10, and advection dominates. Furthermore, the two models

demonstrate that the conductive heat flux contributions may be important with respect to advective and interphase heat flux for sufficiently large initial thermal gradient (here 0.1), while it has been neglected in several earlier investigations (e.g. Schumann, 1929; Spiga and Spiga, 1981).

## 3.2 Maximum temperature difference

The maximum temperature difference of a model can be defined as the maximum value reached in space and time (c.f. Fig.

2f). A series of models has been carried out for the two different non-dimensional parameters $Pe$, and $G = 1/H$, and $\Delta T_{max}$ has been determined for each model (Fig. 3). Some first observations can be made:

- For all $Pe$, $\Delta T_{max}$ is proportional to $Pe$ (Fig. 3a) as long as $\Delta T_{max}$ is somewhat smaller than the absolutely possible maximum 1 which is asymptotically approached for high $Pe$.

- $\Delta T_{max}$ is proportional to $G$, i.e. to the non-dimensional temperature gradient for $G < 0.1$.

- $\Delta T_{max}$ reaches a maximum for large $G$ of order 1, i.e. when $H$ reaches 1 or the dimensional $H$ reaches the length scale $L$.

- $\Delta T_{max}$ is essentially independent of $\phi$ as models with different $\phi$ almost merge in the same points shown in Fig. 3. This has been verified by running all models of Fig. 3 with melt fractions between 0.1 and 0.9 (not shown).

These observations suggest the existence of several domains in which scaling laws for $\Delta T_{max}$ could be derived, based on the

two scaling parameters. In the next section, we propose an analytical derivation of $\Delta T_{max}$ - values to obtain scaling laws and confirm the observed proportionalities.

## 4 Scaling laws derived from analytical solution

In this section a simplified analytical solution for the z-dependent temperature difference between fluid and solid will be derived. From this solution the maximum temperature differences $\Delta T_{max}$ can be obtained and scaling laws will be derived.

**4.1 Analytical solution of the governing equations**

We are interested in an analytical solution of equation (20) controlling the non-equilibrium temperature difference $\left(T_f - T_s\right)$. We simplify the problem by considering the hypothetical case in which $\left(T_f - T_s\right)$ does not change with time, and, moreover, in which the thermal gradient in the solid phase is fixed and linear, with $\partial T_s/\partial z = -G = -1/H$ (non-dimensional, with dimensions: $G = \Delta T_0/H$). Although different from initial and steady state stages described in the 1D models (section 3.1), this

hypothetical case is quite similar to what can be observed at the very beginning of the second stage described in section 3.1 (c.f. Fig. 2f,h). In this second stage, the evolution of $T_f$ and $T_s$ was observed being quite similar indeed. Besides, at the end of stage 1 (section 3.1), $T_s$ remains close to initial conditions, therefore a fixed linear gradient of slope $-G = -1/H$ is justified. Since the maximum temperature difference between the two phases is observed starting from the end of stage 1 and during stage 2 (section 3.2), it does not seem unreasonable to consider this hypothetical case for deriving the maximum temperature

difference. Using these assumptions, Eq. (20) becomes:

$$\frac{\partial^2 (T_f - T_s)}{\partial z^2} - Pe \frac{\partial (T_f - T_s)}{\partial z} - (T_f - T_s) = -Pe\, G \tag{21}$$

While in the general case of Eq. (20) the temperature difference implicitly depends on $\phi_0$, i.e. on the three non-dimensional parameters $Pe$, $\phi_0$, and $G$, Eq. (21) no longer depends on $\phi_0$ because we replaced $\partial T_s(\phi_0)/\partial z$ by $-G$ which is independent of $\phi_0$. Eq. (21) is a second order ordinary differential equation for $(T_f - T_s)$ whose solution can be analytically derived as (see

supplementary material for details)

$$T_f - T_s = \alpha e^{r_1 z} + \beta e^{r_2 z} + PeG , \tag{22}$$

where $r_1$ and $r_2$ are the roots of the associated equation of Eq. (21)

$$r_1 = \frac{1}{2}\left(Pe - \sqrt{Pe^2 + 4}\right), \quad r_2 = \frac{1}{2}\left(Pe + \sqrt{Pe^2 + 4}\right) . \tag{23}$$

The parameters $\alpha$ and $\beta$ are constrained by the boundary conditions: $(T_f - T_s) = 0$ at $z = 0$ and $\frac{\partial (T_f - T_s)}{\partial z} = 0$ at $z = H$

$$\alpha = PeG \frac{r_2}{r_1 e^{(r_1 - r_2)/G} - r_2}, \quad \beta = PeG \frac{r_1}{r_2 e^{(r_2 - r_1)/G} - r_1} . \tag{24}$$

The third term in Eq. (22) is a particular solution for Eq. (21).

## 4.2 Comparison with numerical models

From Eq. (22) the maximum value of the depth-dependent temperature difference $(T_f - T_s)$ can be determined. It is found that the maximum is always at $z = H$. This value will be denoted as $\Delta T_{max}$ and has been calculated for all parameter combinations

used for the numerical models. In Fig. 3 these analytical solutions are plotted as solid lines together with the numerical solutions (asterisks). The agreement is very good, for most cases the differences between the numerical and analytical solutions are well below 1%, only when $\Delta T_{max}$ reaches values of about 0.6 and higher the differences become $> 1\,\%$, up to 6%. This general good agreement is another justification for using the time-independent equation (21) to obtain an analytical solution of an intrinsically time-dependent process as long as we are interested only in the maximum value of $(T_f - T_s)$. Other reasons for

the observed differences between the analytical and numerical solutions include numerical errors when determining the particular times when maximum temperature differences are reached, especially for the models which are in the regime close to $\Delta T_{max} = 1$ where the $\Delta T_{max}(Pe)$ – curves become non-linear (Fig. 3a).

## 4.3 Scaling laws for temperature differences at certain parameter limits

The analytical solution for $\Delta T_{max}$ fits very well with our model results and therefore looks to be ideal for getting a better
understanding of the relative influences of the two controlling parameters $Pe$ and $G$, described in sections 2.2 and 2.3. The Peclet number is already known to be of great importance for thermal equilibrium/non-equilibrium conditions. Inspecting the last term in Eq. (22) we notice that a high $Pe$ and a high initial thermal gradient should favor higher temperature differences. This has been demonstrated in Fig. 3.

Eq. (22) is, however, complicated, and the assessment of the relative importance of $Pe$ and $G$ for different possible regimes is
limited. In this section, we study the evolution of $\left(T_f - T_s\right)$, i.e. also $\Delta T_{max}$, in a few limiting cases. This enables us to better understand the influence of each parameter and to derive some scaling laws for different regimes.

### 4.3.1 Limit $Pe \rightarrow 0$

When $Pe$ tends to 0, we have the condition

$$Pe \ll 2 \tag{25}$$

With this condition Eq. (22) tends to the following limit (see supplementary material):

$$T_f - T_s = PeG(1 - M) \tag{26}$$

with

$$M = \frac{\cosh(z) + \cosh\left(\frac{2}{G} - z\right)}{1 + \cosh(2/G)} \tag{27}$$

which simplifies for $z = H = 1/G$ to

$$M = \frac{1}{\cosh(1/G)} \tag{28}$$

This is the limit for $Pe \rightarrow 0$. This limit gives predictions for $\Delta T_{max}$ in very good agreement with Eq. (22) for $Pe < 1$ (having $G = 0.1$) (see inset in Fig. 3a or Fig. S1 in the supplementary material). In the limit $G \rightarrow 0$ and finite $Pe < 1/G$ we get the limit for $M$

$$M \rightarrow e^{-z}$$

Thus, for both small $Pe$ and small $G$ the temperature difference (Eq. 26) can be written as

$$T_f - T_s = PeG(1 - e^{-z}) \tag{29}$$

Eq. (29) confirms the proportionalities observed in Fig. 3, namely $\Delta T_{max} \propto Pe$ (Fig. 3a), and $\Delta T_{max} \propto G$ (Fig. 1b), respectively.

### 4.3.2 Limit $Pe \rightarrow \infty$

To obtain the limit of Eq. (22) for $Pe \rightarrow \infty$, Eq. (22) can be linearized with respect to $4/Pe^2 \ll 1$. Applying the rule of L'Hôpital Eq. (22) tends to the following limit:

$$T_f - T_s = Gz \tag{30}$$

For details, see supplementary material. This limit is also the solution of Eq. (21) when neglecting the diffusive and heat transfer terms. As demonstrated in the supplementary material this limit predicts $\Delta T_{max}$ - values in very good agreement with Eq. (22) for $Pe > 100$ (Fig. 3a, inset).

### 4.3.4 Exploring the domains for the maximum temperature difference including all limits

Before exploring the full parameter space we first give a short overview of expected parameter ranges in magmatic systems. In natural magmatic systems such as mid-ocean ridges, $Pe$ is expected to evolve from very low values of order $10^{-5}$ to $10^{-3}$ in partially molten regions with distributed porous flow to higher values of order 1 or larger at depths where channels have merged, and further to very high values of order $10^5$ in dyke systems (Schmeling et al., 2018).

While the melt fraction does not influence $\Delta T_{max}$ (c.f. Eq. (22, 30)) it influences the long term temporal behavior because $T_s$ is $\phi_0$ – dependent (c.f. Eq. (20)). Therefore, some words about possible melt fractions. As melt flow may occur at very small melt fractions (McKenzie, 2000; Landwehr et al., 2001), large $\phi$ - values are not expected in natural mantle magmatic systems, nor in dyke systems in the crust. Values of channel volume fraction generally remain below a few percent up to tens of percent (in dunite channels up to 10 - 20%, Kelemen et al., 1997).

To get an idea about the expected order of magnitude of the macroscopic thermal gradient $G = 1/H$ of the system we have to evaluate the scaling length $L$ used to scale the dimensional $H$. $L$ scales with the geometric mean of the channel width $d_f$ and the interfacial boundary layer thickness $\delta$ (Eq. 9 with 11). $L$ would evolve non-linearly with the width of melt pathways, which may increase by several orders of magnitude as 3D grain junctions eventually merge to 1D dykes. As will be shown in section 5.3 in more detail the resulting non-dimensional $G$ ranges between order 1 to order $10^{-5}$.

In Figure 4 we explore $\Delta T_{max}$ variations using the analytical solution Eq. (22), in which $\Delta T_{max}$ depends on $Pe$ and $G$. Three main regimes can be distinguished:

- Regime 1: For high $Pe$ - values, $\left(T_f - T_s\right)$ tends to the relationship described in Eq. (30). The temperature difference increases linearly with distance from the bottom (z = 0) reaching $\Delta T_{max} = 1$ at $z = H$. In the whole region the fluid temperature remains constant and at maximum 1 while the solid temperature increases linearly with z from 0 to 1. The proportionality of $\Delta T_{max}$ to $G$ disappears because the maximum value of $z$ is equal to $H = 1/G$.

- Regime 2: For $Pe \ll 1$, or more precisely, for $Pe \ll \frac{1}{G}$ represented by the oblique dashed line in Fig. 4, $\left(T_f - T_s\right)$ varies with distance from the bottom according to $(1 - e^{-z})$, and is proportional to $Pe$ and $G$. This means that large temperature gradients favor large temperature differences. In this domain, $\left(T_f - T_s\right)$ tends to the relationship presented in Eq. (29).

- Regime 3: For large initial temperature gradient $G$ close to 1 (small $H$) and $Pe \ll 1$, $\left(T_f - T_s\right)$ tends to the relationship proposed in Eq. (26). In this domain, $\left(T_f - T_s\right)$ is proportional to $Pe$ but no more to $G$ because $M$ is a function of $G$ gradually canceling the proportionality to $G$, which is visible in regime 2. The depth-dependence is given by $\left(1 - M(z)\right)$ which at $G = 1$ increases non-linearly from about 0 to 0.4 with increasing z.

## 5 Discussion

### 5.1 Limitations

#### 5.1.1 Comments on the analytic solution

Although the assumptions used to get the analytic solution (Eq. 22) are very specific, they are reasonable considering the conditions in the models when $\Delta T_{max}$ is reached, and it fits very well the numerical results. This is shown in Fig. 5 where for various combinations of $Pe$ and $G$ the time-dependent temperature differences $(T_f - T_s)$ are shown as functions of depth together with the analytical solutions using Eq. (22). For all examples the position of the maximum temperature differences lies at $z = H$. A major simplification used in Eq. (21) was time-independence. Obviously, the resulting analytical solutions represent the stage 2, which is quasi steady state in contrast to stage 1 when the temperature difference builds up, and stage 3 when the long-term behavior is approached. We emphasize that this analytical solution is a very good approximation of the depth-dependent temporal maximum temperature difference that can be reached in such porous systems.

#### 5.1.2 Boundary conditions at top and initial conditions

The boundary conditions we chose at the top ($z = H$) are suitable for cases with little temperature evolution (regime 2 and 3, low $Pe$), and for early stages for regime 1 but might be inappropriate for high temperature increases (high $Pe$ – regime 1) at later stages (see section 4.3.4). In order to quantify the influence of this choice of boundary conditions on our results, we compared the evolution of $(T_f - T_s)$ - profiles for three Peclet numbers and two values of $G$, using four different boundary conditions at the top (Fig. 6):

- Constant thermal gradient equal to the initial thermal gradient in the solid and fluid phases (Neumann condition). This was the boundary condition used in the models.

- Thermal gradient is set to 0 at the top (Neumann condition).

- Both fluid and solid temperatures are set to 0 at the top (Dirichlet condition).

- Temperature at the top is numerically calculated from the full equations (17) and (18) using one-sided (upwind) positions for the first and second derivatives (open boundary).

Mathematically, the open boundary condition is not a rigorous boundary condition because both the temperature and temperature gradient intrinsically depend on the temperature evolution within the model. Therefore, it cannot be applied to the analytical solution of section 4.1. Numerically it works well for our system without producing instabilities or oscillations. Comparing the top and bottom row of Fig. 6, the constant temperature gradient condition produces quite similar results as the open boundary condition for all $Pe$ and $G$ - values tested during the first and second stage of temporal evolution (c.f. section 3.1). The agreement becomes worse for stage 3 when approaching steady state at large $Pe$. Comparing the other two boundary conditions (2nd and 3rd row of Fig. 6) with the constant gradient condition (top row) shows that the effect of the top boundary during stage 1 and 2 is still small sufficiently far away from the top. Only for the small $Pe$ - case (left column of Fig. 6) the

zero gradient and zero temperature conditions strongly affect the upper half of the domain by diffusion. Yet the maximum temperature difference of the constant gradient case is nearly reached by the other two boundary conditions further within the domain, not at the top. The special case of high $Pe$ and small $G$ with zero temperature boundary condition (3$^{rd}$ row 4$^{th}$ column in Fig. 6) shows a strong build-up of $T_f - T_s$ close to the top when approaching the steady state. This stems from the large local temperature gradient built up near the top as a result of transforming the difference in advective heat in- and output $\left(Pe\, T_{influx} - Pe\, T_{outflux} = Pe\right)$ into a high conductive outflux $(\partial T / \partial z)$ at the top. It is unlikely that such situations occur in natural systems.

We have also tested an open boundary condition for the fluid and a Robin boundary condition for the solid imagining a lid on top of our model with a constant temperature gradient and a fixed surface temperature. Choosing the surface temperature in such a way that the initial thermal gradient within our system and within the lid are identical, this boundary condition adds a new non-dimensional parameter, the lid thickness $H_{lid}$. Tests show that lid thicknesses larger than $H$ give results in general agreement with the constant gradient or open boundary models of Fig. 6. Localized differences with up to 30 % higher $\left(T_f - T_s\right)$ – values occur near the top, but they disappear for $H_{lid} \geq 10\, H$. Lid thicknesses smaller than $H$ force the solid temperature at the top to values closer to 0, while the fluid temperature remains high. This results in significantly higher $\left(T_f - T_s\right)$ – values than in Fig. 6 (top row) close to the top. However, typical natural magmatic systems are on the $H_{lid} > H$ side suggesting that our constant gradient boundary condition is a good approximation.

In summary, the influence of boundary conditions on fluid and solid temperature evolution depends mostly on the domain size $H$ and on the value of $Pe$. The larger these two parameters, the less important is the influence of boundary conditions within almost the whole model domain. If one is interested in the maximum value of $T_f - T_s$ in space and time, the tests show that this value can safely be picked at $z = H$ when using the constant temperature gradient boundary condition.

As an initial condition we used a linear temperature profile and initial equilibrium between solid and fluid. A non-linear initial temperature profile between $T_f = T_s = 1$ at the bottom and $T_f = T_s = 0$ at the top would have spatially varying temperature gradients with sections with gradients larger than those assumed in our model. As the temperature gradient strongly influences thermal non-equilibrium (see e.g. Eq. 22 which explicitly contains the temperature gradient $G$), the above results are expected to be different, and a stronger thermal non-equilibrium is expected in regions with higher gradients. Schmeling et al. (2018) used a step function with $T_f = T_s = 1$ at $z = 0$ and $T_f = T_s = 0$ at $z > 0$ as initial condition, i.e. an extremely non-linear profile near $z = 0$. Assuming this initial temperature profile Figure 7 shows the temporal behavior of the temperature difference for selected parameter combinations, equal to the parameters used in Fig. 5. The analytical solutions for the time-independent cases (Eq. 22) are also shown. As expected, at early stages the temperature differences are significantly larger than given by the analytical solutions by a factor 2 or more shortly after the onset of the evolution. At later stages (stage 2 or 3) the time-dependent solutions approach or pass through the analytical solutions. Thus, we may state that the analytical solutions depicted in the regime diagram in Fig. 4 represent lower bounds of thermal non-equilibrium compared to settings with non-linear initial temperature profiles.

### 5.1.3 Different densities and thermal properties of the two phases

While for simplicity, we used equal physical properties for the fluid and solid, in many circumstances they might be significantly different. Equal properties are good approximations for magmatic systems where differences of density and thermal parameters are small (order of 10%), whereas porous flows of water or gases through rocks or other technical settings may be characterized by larger differences. Allowing for different material properties adds four new parameters, namely the ratio of diffusivities, the ratio of densities, the ratio of heat capacities and a new effective thermal conductivity $\lambda_{eff}$ for the

interface between the two phases with different properties. To evaluate how many new non-dimensional numbers are introduced we non-dimensionalize the equations assuming different material properties for the two phases. We use the fluid properties as scaling quantities and assume that they are independent of temperature, pressure and depth. We modify the scaling length and Peclet number by defining $\tilde{L} = \frac{L}{\sqrt{\lambda'_{eff}}}$ with $\lambda'_{eff} = \lambda_{eff}/\lambda_f$ and $\widetilde{Pe} = \frac{Pe}{\sqrt{\lambda'_{eff}}}$. With this scaling eq. (14) and (15) turn into (for clarity, primes indicate non-dimensional quantities):

$$\phi\left(\frac{\partial T_f'}{\partial t'} + \widetilde{Pe}v' \cdot \nabla T_f'\right) = \nabla \cdot (\phi \nabla T'_f) - \phi_0(1 - \phi_0)(T_f' - T_s') \tag{32}$$

and

$$(1 - \phi)\frac{\partial T_s'}{\partial t'} = \frac{\kappa_s'}{\rho_s' c_{p,s}'} \nabla \cdot ((1 - \phi)\nabla T_s') + \phi_0(1 - \phi_0)\frac{1}{\rho_s' c_{p,s}'}(T_f' - T_s') \tag{33}$$

Inspection of these equations shows that two more non-dimensional numbers are introduced: the ratio of diffusivities $\kappa_s'$ and the ratio of the products density and heat capacity, $\rho_s' c_{p,s}'$.

As equations (32) and (33) cannot be merged into one time-independent ordinary differential equation for $(T_f - T_s)$ as in section 4.1, we numerically tested some cases with $\widetilde{Pe} = Pe = 1$, i.e. and $\lambda_{eff}' = 1$ in which the diffusivity ratio and the ratio of $\rho_s' c_{p,s}'$ were varied between 0.1 and 10 (see Fig. 8). The results show that for the fixed combination of $Pe = 1$ and $\lambda_{eff}' = 1$ the magnitude of thermal non-equilibrium remains in the same order of magnitude $O(0.1)$ as for equal properties (Fig. 8). However, the time-dependence is significantly affected: For a high ratio of $\kappa_s' = 10$ (i.e. the solid is strongly conducting) the

solid temperature profile remains close to the constant initial gradient, and the temperature difference rapidly converges to a steady state similar to the analytical solution depicted in Fig. 5a. In contrast, for a low $\kappa_s' = 0.1$ the solid temperature departs more strongly from the initial linear gradient, and the solid – fluid temperature difference slowly drops with time on the long term. Varying the potential to store heat in the solid, i.e. $\rho_s' c_{p,s}'$, Fig. 8e and f shows that a high value slows down the long term time-dependent variations, while a small value leads to rapid long term temporal variations of $(T_f - T_s)$ and faster

convergence to the steady state which is similar to the equal properties case.

It is interesting to apply the results for different physical properties to a geologically relevant setting, namely water flowing through sedimentary rocks. Given that the high heat capacity of water is about three times larger than that of rock, and the density is almost three times less, the product $\rho_s' c_{p,s}'$ is about 0.78, i.e. of order 1. However, the thermal diffusivity of water is significantly smaller than that of rock, typically by a factor 16, i.e. $\kappa_s'$ is about 16. We tested a few cases (Fig. 9) with Peclet

numbers and initial thermal gradients $G$ (i.e. inverse model heights) (assuming for simplicity $\lambda_{eff}' = 1$) equal to the cases depicted in Fig. 5. The time dependent profiles behave similarly to those in Fig. 5, with very similar maxima of the temperature differences (red dashed curves in Fig. 5) relevant for stage 2. The only important difference is that the water-sedimentary rock case more rapidly approaches the late steady states of stage 3 and these stages are closer to the maximum red-dashed curves. These results suggest that the absolute values of maximum thermal non-equilibrium temperature differences shown in the

regime diagram Fig. 4 are also applicable to a water-sedimentary rock system.

## 5.2 Time scales

It is interesting to evaluate the time scales for reaching the maximum non-equilibrium temperature differences and the steady state. For every numerical model, we recorded the time needed to reach 90% of the maximum temperature differences between fluid and solid, $t_{90\%}$, and the time needed to reach steady state, $t_{steady}$. The latter has been determined as the time at which

the maximum difference between $\left(T_f(z) - T_s(z)\right)$ – curves at two subsequent time steps becomes less than $10^{-8}\Delta T_{max}$. These times can be compared with different time scales that may characterize the evolution of temperatures in the models. These time scales can be based on advection over a characteristic distance $d_{char}$ giving $t_{advd} = d_{char}/v_{f0}$, or on diffusion over the characteristic distance giving $t_{diffd} = d_{char}^2/\kappa_0$. We tested these time scales with the two natural length scales of the models. The first is the scaling length $L$ (= 1 non-dimensional) representing essentially the geometric mean of the channel width of the

pores, $d_f$, and the interfacial boundary layer thickness $\delta$. The second is the model height $H$. Grouping the models depending on the regime they belong to (see section 4.3.4, and Fig. 4), we plotted the recorded times $t_{90\%}$ and $t_{steady}$ versus the characteristic time scales mentioned above. Good agreement with the characteristic time scales is indicated by observed times fitting to the dashed x = y - lines (Figure 10).

- In regime 1 (high $Pe$), $t_{90\%}$ is proportional to $t_{advH}$ (Figure 10a, blue circles). In this regime the high value of $Pe$
makes the fluid temperature increase fast. It reaches its maximum value during the time under which significant fluid-solid heat transfer builds up and the solid temperature is still low. This corresponds to the time for traveling the full distance $H$. During stage 2 and 3 the solid temperature increases and the temperature difference decreases before steady state is reached. The time for reaching steady state (Fig. 10b, circles) varies roughly linearly with $t_{steady} \propto$ $t_{diffH}$. For most cases it is controlled by diffusion through the solid over distances of order $H$. The case with large $H$
(circle in Fig. 10b below dashed line) apparently reaches the steady state earlier, but still later than on a corresponding advective time scale based on $H$ (not shown). Inspecting this model shows that during stage 2 and 3 the high $Pe$ number facilitates approaching thermal equilibrium rapidly within large parts of the model and reducing the effective length scale (and characteristic timescale) over which still non-equilibrium is present.

- In regime 2 (low $Pe$ and $G < 0.1$, i.e. $H > 10$) the time for reaching $\Delta T_{max}$ is controlled by interfacial heat transfer
(Fig. 10a, red asterisks) on the length scale $L$ resulting in $t_{90\%}$ proportional to $t_0$. The time for reaching steady state is controlled by the diffusion time scale across the height of the system (Fig. 10b).

- In regime 3, (low $Pe$ and high $G$ (small $H$)), time for reaching $\Delta T_{max}$ is similar or shorter than the diffusion time based on the model height $H$ (Fig. 10a, black crosses). The flattening of the curve indicates that non-equilibrium is reached faster for some models because $Pe$ reaches order 1 and the advective timescale starts to take over. The time for reaching steady state (Fig. 10b, crosses) varies linearly with $t_{steady} \propto t_{diffH}$. Clearly, it is also controlled by diffusion.

## 5.3 Applications to magmatic systems

We now test the possible occurrence of thermal non-equilibrium in natural magmatic systems based on the suggested controlling non-dimensional parameters, namely the Peclet number $Pe$, the initial thermal gradient $G$ ($= 1/H$), and the melt fraction $\phi$. Typical stages of melt flow for mid-ocean ridges include three stages:

a) partially molten regions with interstitial melts sitting at grain corners, grain edges or grain faces with low (0.0001 - 6%) melt fractions (see e.g. the discussion in Schmeling, 2006),

b) merging melt channel or vein systems with high (> 10 - 20%) porosity channels identified as dunite channels after complete melt extraction (Kelemen et al., 1997),

c) propagating dykes or other volcanic conduits.

Let's assume typical overall melt fractions of 1% to 20% for stages $b$ and $c$. Schmeling et al. (2018) discussed possible Peclet numbers for such systems based on a Darcy flow related Peclet number

$$Pe_D = \frac{v_D d_s}{\kappa_0} \tag{34}$$

As we preferably use the melt pore dimension $d_f$ in our scalings (Eq. 9a and 10a) we need to relate it to the solid phase dimension $d_s$ by using

$$d_s = d_f \frac{g}{\phi}, \quad g = \begin{cases} (1-\phi) & \text{melt channels} \\ \sqrt{\phi}(1-\sqrt{\phi}) & \text{melt tubes} \end{cases} \tag{35}$$

Using (35), (9a), and (16) we arrive at the Peclet number used here

$$Pe = Pe_D \frac{1}{g\sqrt{c}} \sqrt{\frac{(1-\phi)\delta}{d_f}} \tag{36}$$

Schmeling et al. (2018) reviewed and estimated typical pore or channel spacings $d_s$ of $10^{-3} - 10^{-2}$ m for stage $a$, 0.1 m for early stage $b$ increasing to 1 -100 m for late stage $b$, and 100m – 300 m for stage $c$ (dykes). Arguing for typical geometries, spreading rates and melt extraction rates Schmeling et al. (2018) estimated the Darcy velocity ranging between $10^{-10}$ m/s and $10^{-9}$ m/s. With these parameters $Pe_D$- numbers for the three stages can be estimated for the tree stages as

a) $10^{-7}$ to $10^{-5}$,

b) $10^{-5}$ to $10^{-4}$ at depths where channel distances are of order 0.1 m, and $10^{-4}$ to 0.1 at shallower depths where the channel distances have increased to the order of 1m to 100 m,

c) $>10^5$ for the dyke stage.

To estimate Peclet numbers as defined here (Eq. 36) typical interfacial thermal boundary layer thicknesses $\delta$ are needed. As the thermal interfacial heat exchange intrinsically is time-dependent a good estimate is $\delta = c_{th}\sqrt{\kappa_0 t}$ (in dimensional form) where $c_{th}$ is a constant for a thermal boundary layer, equal to 2.32 for a cooling half space. Assuming that the characteristic time can be expressed by the (dimensional) fluid velocity $v_0$ and system height $H$, i.e. by $t = H/v_0 = H\phi/v_D$, we may express $v_D$ in terms of the Peclet number $Pe_D$. With the resulting $t$ and $\delta$ we arrive at the following Peclet number ($H$ and $d_f$ are dimensional or non-dimensional):

$$Pe = Pe_D{}^{3/4}\sqrt{\frac{c_{th}}{c}}g^{-3/4}\left(\frac{H}{d_f}\right)^{1/4}\sqrt{1-\phi} \qquad (37)$$

For mid-ocean ridge settings we assume $H$ representing the transition region between the lithosphere and asthenosphere with a thickness of the order 1 to 10 km, and use Eq. (35) to insert typical $d_f$ – values. They increase from $10^{-4}$ m for interstitial melts (stage $a$) to $10^{-3}$ m to $10^{-1}$ m for the channeling stage $b$ (see Schmeling et al., 2018) to >10 m for the dyke stage $c$. The resulting Peclet number (Eq. 37) is of the order $10^{-3}$ to 0.5 for stage $a$, order $10^{-2}$ during the early stage $b$ and order $10^{-2}$ to 1 during the later stage $b$ appropriate for dunite systems, and order $10^4$ to $10^7$ for the dyke stage $c$. To estimate typical non-dimensional thermal gradients $G'$ (or layer thickness $H'$) the above estimate for $\delta$ and $d_f$ can be inserted into the scaling length $L$ (Eq. 9a) to arrive at a non-dimensional $G'=1/H'$

$$G' = \left(\frac{H}{d_s}\right)^{-3/4}g^{-1/2}\phi^{3/4}\sqrt{\frac{c_{th}}{c}}Pe_D{}^{-1/4}\sqrt{1-\phi} \qquad (38)$$

With the derived estimates for the three stages, $G'$ is of the order $10^{-6}$ to $10^{-2.7}$ for stage $a$, $10^{-4} – 10^{-2.5}$ increasing to $10^{-4} - 0.6$ for stage $b$, and $10^{-5} – 10^{-2}$ for the dyke stage $c$. These resulting stages for $Pe$ and $G'$ are indicated in the regime diagram (Fig. 4). All three stages extend far into the domain with $G$ - values smaller than 0.001. Thermal non-equilibrium of the three stages can be summarized:

a) Interstitial melts are at *full thermal equilibrium*,

b) channeling and veining may result in *moderate thermal non-equilibrium* at sufficiently high thermal gradients,

c) after transition to dyking *full thermal non-equilibrium* is predicted.

A similar exercise can be done for continental magmatic systems. We skip such an explicit evaluation here but note that silicic melt viscosities are typically higher than those of basaltic melts at mid-ocean ridges. Thus, Peclet numbers are expected to be smaller, but non-dimensional thermal gradients (Eq. 38) might be larger, resulting in a downward and rightward shift of the natural stages indicated in Figure 4.

To make our scaling laws and time scales for reaching maximum thermal non-equilibrium more accessible it is worth writing them in dimensional form. First, to estimate the Peclet number of a natural system combining Eq. (9) and (16) gives

$$Pe = \frac{v_{f0}}{\kappa_0}\sqrt{\frac{\phi_0(1-\phi_0)\delta}{S}} \qquad (39)$$

indicating that for very small or very large melt fractions $Pe$ becomes very small. One may use Eq. (11) or (12) to write $Pe$ also in terms of pore or solid (grain or channel spacing) dimensions $d_f$ or $d_s$, respectively. The scaling laws and characteristic time scales for the three regimes we found (Fig. 4) are in dimensional form:

- Regime 1: For large $Pe$ the maximum non-equilibrium temperature difference is simply equal to the imposed temperature difference, $\Delta T_{max} = \Delta T_0$, and the characteristic time to reach maximum non-equilibrium is simply $t_{char} = H/v_{f0}$, i.e. the total time of a fluid particle for passing through the system.

- Regime 2 and 3: For small Peclet number ( $Pe < \frac{H\sqrt{S}}{\sqrt{\phi_0(1-\phi_0)\delta}}$) the maximum temperature difference scales like

$$\Delta T_{max} = \frac{Gv_{f0}\phi_0(1-\phi_0)\delta}{\kappa_0 S} = \frac{v_{f0}\phi_0(1-\phi_0)\delta}{H\kappa_0 S}\Delta T_0 \tag{40}$$

and the characteristic time for reaching this non-equilibrium scales with $t_0$, i.e.

$$t_{char} = \frac{\phi_0(1-\phi_0)\delta}{\kappa_0 S} \tag{41}$$

These relations can easily be used to assess the potential of thermal non-equilibrium in systems of fluid flow through solids with given geometrical properties and fluid fractions.

In the above discussion we used the terms *moderate thermal non-equilibrium* which we may identify with $\Delta T'_{max}$ of a few % to, say, 30%, while *full thermal non-equilibrium* includes higher values up to 100%. To translate this into dimensional $\Delta T_{max}$ - values, what are typical $\Delta T_0$ - values for mid-ocean ridges? In our example we defined $H$ as the thickness of the transition region between lithosphere and asthenosphere. Such a transition zone may be defined by the depth region bounded by the asthenospheric temperature $T_{asth}$ and, say, $0.8T_{asth}$, i.e. $\Delta T_0 = 0.2T_{asth} \cong 200K$. Thus moderate non-equilibrium may be represented by excess temperatures of the melt with respect to the solid between, say, 6 K and 60 K, while full thermal non-equilibrium suggests the full $\Delta T_0$ – range of 60 K to 200 K or even higher in case of dykes extending up into the lithosphere with temperatures below $0.8T_{asth}$. These typical temperature estimates may have some implications for whether the solid rock will melt or the melt will freeze. However, this discussion is beyond the scope of this paper.

**6 Conclusions**

In conclusion, we showed that in magmatic systems characterized by two-phase flows of melts with respect to solid, thermal-non-equilibrium between melt and solid may arise and becomes important under certain conditions. The main conclusions are summarized as follows:

From non-dimensionalization of the governing equations three non-dimensional numbers can be identified controlling thermal non-equilibrium: the Peclet number $Pe$, the melt porosity $\phi$, and the initial non-dimensional temperature gradient $G$ in the system. The maximum possible non-equilibrium solid – fluid temperature difference $\Delta T_{max}$ is controlled only by two non-dimensional numbers: $Pe$ and $G$. Both numerical and analytical solutions show that in a $Pe - G$ - parameter space three regimes can be identified:

- In regime 1 (high $Pe$ $(>1/G)$) strong thermal non-equilibrium develops independently of $Pe$, and a non-dimensional scaling law $T_f - T_s = Gz$ has been derived.
- In regime 2 (low $Pe$ $(<1/G)$ and low $G$ $(<0.3)$) non-equilibrium decreases proportionally to decreasing $Pe$ and $G$, and the non-dimensional scaling law reads $T_f - T_s = Pe\, G(1 - e^{-z})$.

- In regime 3 (low $Pe$ $(<1)$ and $G$ of order 1) non-equilibrium scales with $Pe$ and $G$ and is depth-dependent, the scaling law is $T_f - T_s = Pe\, G\left(1 - M(z)\right)$ where $M(z)$ depends on $G$.

Further conclusions include:
- The time scales for reaching thermal non-equilibrium scale with the advective time-scale in the high $Pe$-regime and with the interfacial diffusion time in the other two low $Pe$ number regimes.

- Applying the results to natural magmatic systems such as mid-ocean ridges can be done by estimating appropriate orders of $Pe$ and $G$. Plotting such typical ranges in the $Pe$ - $G$ regime diagram reveals that a) interstitial melt flow is in thermal equilibrium, b) melt channeling as e.g. revealed by dunite channels may reach moderate thermal non-equilibrium, and c) the dyke regime is at full thermal non-equilibrium.
- In the studied setup $G$ was constant leading to conservative estimates of thermal non-equilibrium. Any other depth-

dependent initial temperature distributions generate higher non-equilibrium than reported here.
- The derived scaling laws for thermal non-equilibrium are valid for equal solid and fluid properties. Assuming different properties such as for a water – sandstone system results in similar maximum non-equilibrium temperature differences, but in significantly different time evolutions.

While for simplicity the presented approach has been done essentially for constant model parameters, it can easily be extended
to vertically varying parameters. Thus, tools are provided for evaluating the transition from thermal equilibrium to non-equilibrium for anastomosing systems (Hart, 1993; Chevalier and Schmeling, in prep.).

## 7 Code availability

The MATLAB code is listed in the supplementary material and is available on request.

## 8 Author contribution

The authors contributed equally to this work.

## 9 Competing interests

The authors declare that they have no conflict of interest.

## 10 Acknowledgements

We gratefully acknowledge the excellent reviews by John Rudge and Cian Wilson, who stimulated us into significantly 610 improving the scaling. We acknowledge funding support by the Deutsche Forschungsgemeinschaft (DFG) with the grant no. 403710316.

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

| Symbol | Definition | Units |
|---|---|---|
| $c_{p,f,s,0}$ | Specific heat at constant pressure for the fluid, solid, or reference, respectively | J kg$^{-1}$K$^{-1}$ |
| $c, c_s$ | Geometrical constant for fluid pore space or solid phase, respectively. For melt channels or low porosity films $c = 2$, for tubes $c = 4$ (Eq. 11, 12) | - |
| $c_{th}$ | Constant for thermal boundary layer, 2.32 for cooling half space | - |
| $d_s, d_f$ | Characteristic length scale of solid or fluid phase, respectively | m |
| $f$ | Subscript used for fluid | - |
| $g$ | Function describing part of the $\phi$- dependence of $d_f, d_s$ (Eq. 35) | - |
| $G$ | Initial temperature gradient, taken positive for temperature decreasing with height, mostly non-dimensional | (T m$^{-1}$) |
| $H$ | Height of the model, mostly non-dimensional | (m) |
| $L$ | Scaling length used for non-dimensionalization (Eq. 9) | m |
| $M(z)$ | Function describing the depth-dependence of the analytical solution of $\left(T_f - T_s\right)$ for small $Pe$ (Eq. 27) | - |
| $Pe, Pe_D$ | Peclet number based on fluid velocity (Eq. 16) or based on Darcy velocity (Eq. 34), respectively | - |
| $Q_{fs}$ | Interfacial heat exchange rate from fluid to solid | J s$^{-1}$ m$^{-3}$ |
| $r_1, r_2$ | Constants of analytical solution (Eq. 23) | - |

| | | |
|---|---|---|
| $s$ | Subscript used for solid | - |
| $S$ | Interfacial area density, i.e. interfacial area per volume | m$^{-1}$ |
| $t, t_{char}$ | Time, characteristic timescales, respectively. "char" indicates the characteristic time for diffusion or advection over a characteristic length $L$ or $H$: "diffL", "diffH", "advL", "advH" | s |
| $t_0$ | Scaling time (Eq. 10) | s |
| $T_{f,s}$ | Temperature of the fluid or solid, respectively | K |
| $\Delta T_0, \Delta T_{max}$ | Initial temperature difference between top and bottom used as scaling temperature, and maximum difference between fluid and solid temperature in space and time, respectively | K |
| $v_{f,s}$ | Velocity of the fluid or solid, respectively | m s$^{-1}$ |
| $v_{f0}$ | Constant fluid velocity in the model, used for scaling | m s$^{-1}$ |
| $v_D$ | Volumetric flow rate (Darcy velocity) $(= \phi v_f)$ | m s$^{-1}$ |
| $x, y, z$ | Coordinates, distance | m |
| $\alpha, \beta$ | Functions used for analytical solution (Eq. 24) | - |
| $\delta$ | Interfacial boundary layer thickness | m |
| $\kappa_{f,s,0}$ | Thermal diffusivity of the fluid, solid or reference, respectively | m$^2$ s$^{-1}$ |
| $\lambda_{f,s}$ | Thermal conductivity of the fluid or solid, respectively | W m$^{-1}$ K$^{-1}$ |
| $\lambda_{eff}$ | Effective thermal conductivity at the solid-fluid interface | W m$^{-1}$ K$^{-1}$ |

| $\phi, \phi_0$ | Porosity or scaling porosity, respectively | - |
| $\rho_{f,s,0}$ | Density of the fluid, solid, or reference, respectively | kg m$^{-3}$ |

**Table 1: Symbols, their definition, and physical units used in this study.**







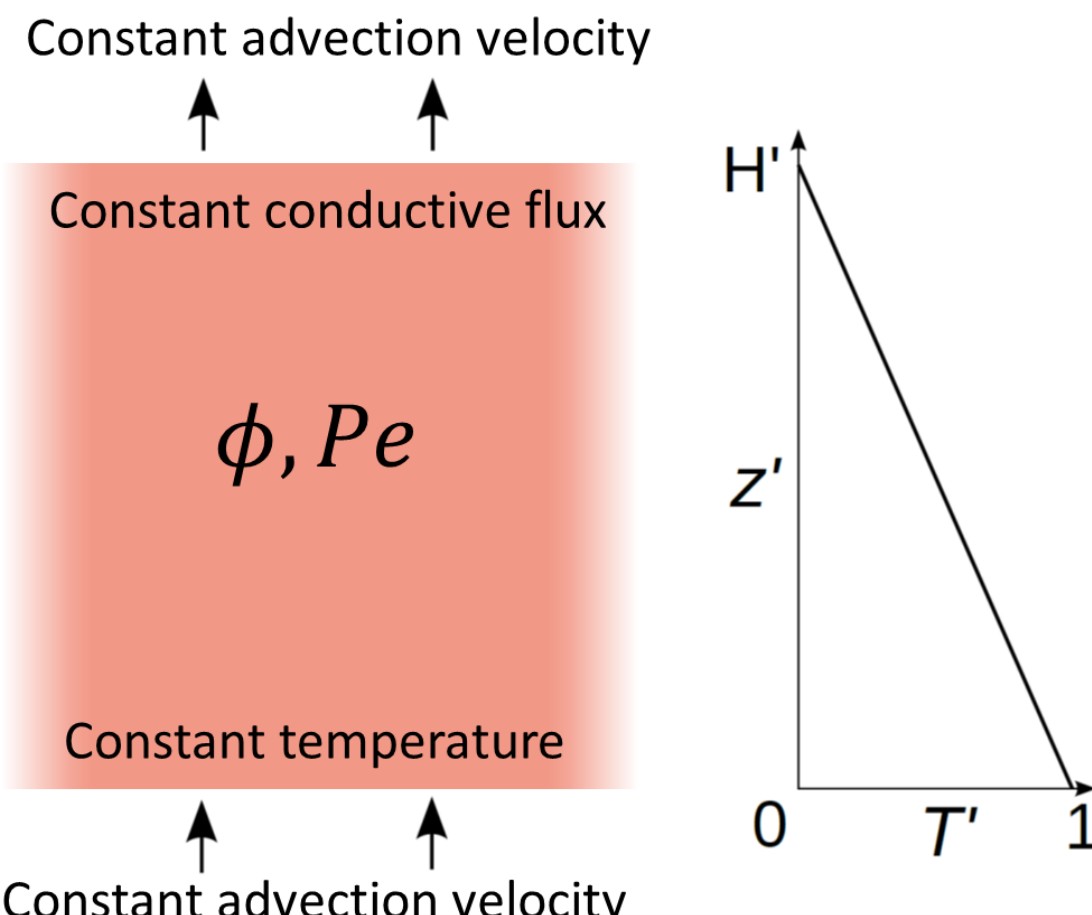

**Figure 1. Initial and boundary conditions.**

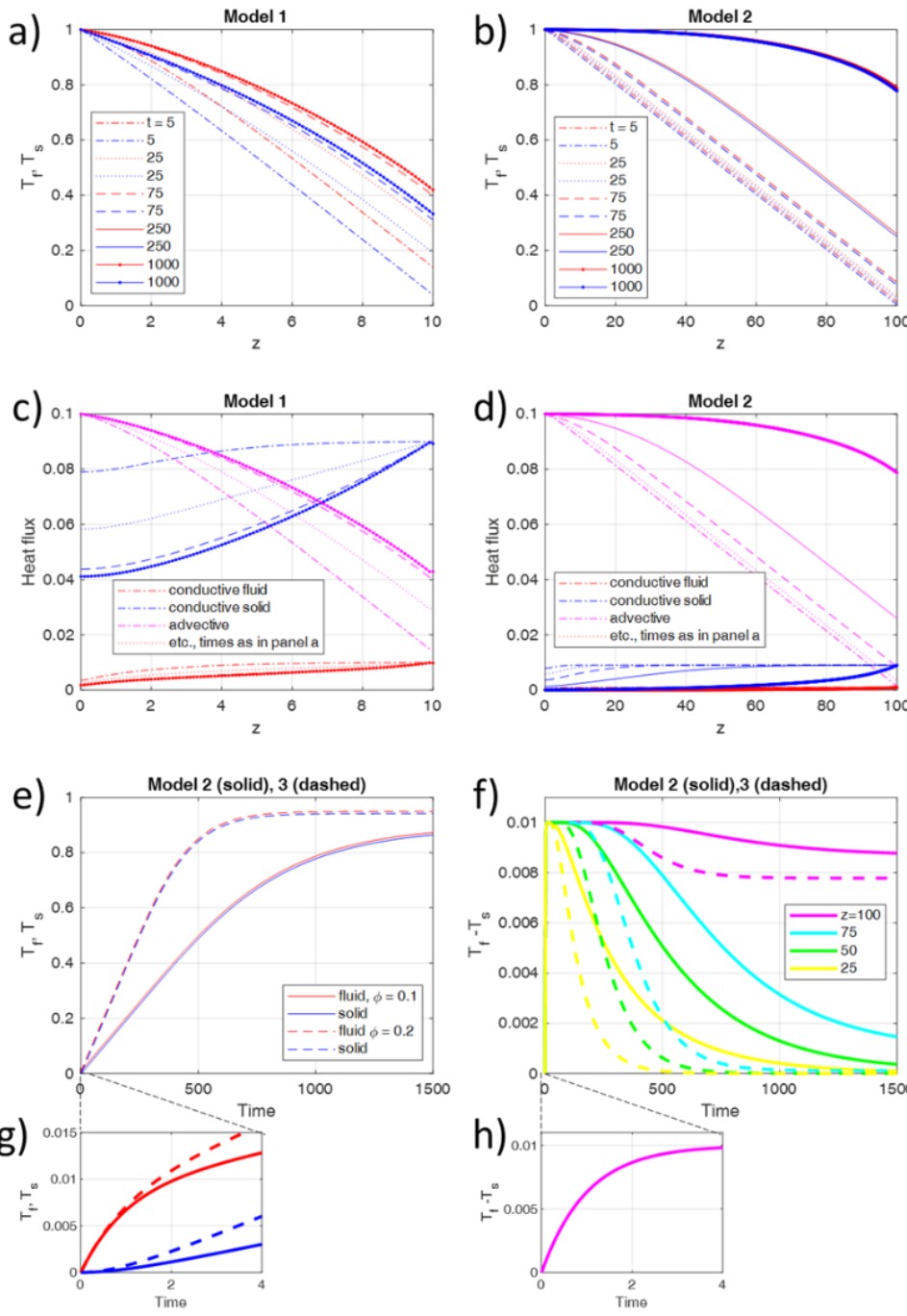

**Figure 2. Typical model evolution for** $Pe = 1$**, two different melt fractions** $\phi$**, and two different non-dimensional temperature gradients** $G$ **(i.e. heights** $H$**). a) Model 1 is with** $G = 0.1$ **(** $H = 10$**) and** $\phi = 0.1$**. Red and blue curves show the fluid and solid temperatures at different non-dimensional times** $t$ **as indicated by the legend, respectively. Initial temperatures are almost identical to the t = 0.5 curves. Steady state is reached at about** $t = 100$**, the curves of the last two times plot on each other. b) Model 2 with** $G = 0.01$ **(** $H = 100$**), else as in a). Steady state is not fully reached. c) Conductive (blue and red curves) and advective (magenta curve) heat fluxes through the solid and fluid, respectively, of model 1. Line styles indicate the same times as in a). d) Same as c) but for model 2. e) Temporal evolution of fluid and solid temperatures,** $T_f$ **(red) and** $T_s$ **(blue), respectively, at the top of model 2 with** $\phi = 0.1$ **and model 3 with** $\phi = 0.2$**.** $G = 0.01$ **(** $H = 100$**) for both models. f) Evolution of fluid - solid temperature difference (** $T_f$ **-** $T_s$**) at different distances** $z$ **in model 2 (** $\phi = 0.1$**, solid curves) and in model 3 (** $\phi = 0.2$**, dashed curves). g) Zoomed-in early temporal evolution of solid and fluid temperatures of models 2 and 3 shown in e). h) Zoomed-in early temporal evolution of temperature difference of model 2 and 3 shown in f).**

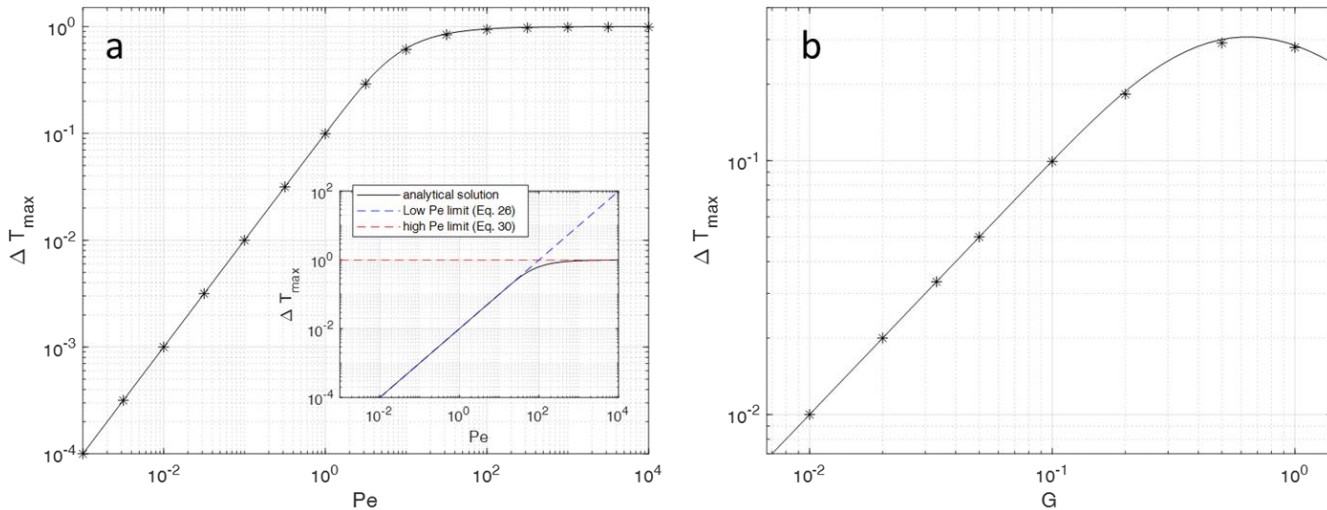

**Figure 3. Maximum fluid – solid temperature differences $T_f - T_s$ of numerical models (asterisks) with different parameters, plotted a) as a function of the Peclet number $Pe$ for $G = 0.1$ and $\phi = 0.1$, and b) as a function of the initial thermal gradient $G$ for $Pe = 1$ and $\phi = 0.1$. The solid lines give the analytic solutions. The inset in a) shows the comparison of the analytic solution Eq. (22) with the different limits derived in section 2.1 and 2.2. The black curve represents the analytic solution, the colored straight lines show the results in the high or low value limits of Eq. (26) to (30), respectively. A larger version of the inset is given as Fig. S1 in the supplementary material. The analytical solution for the full parameter range $Pe - G$ is given in Fig. 4.**

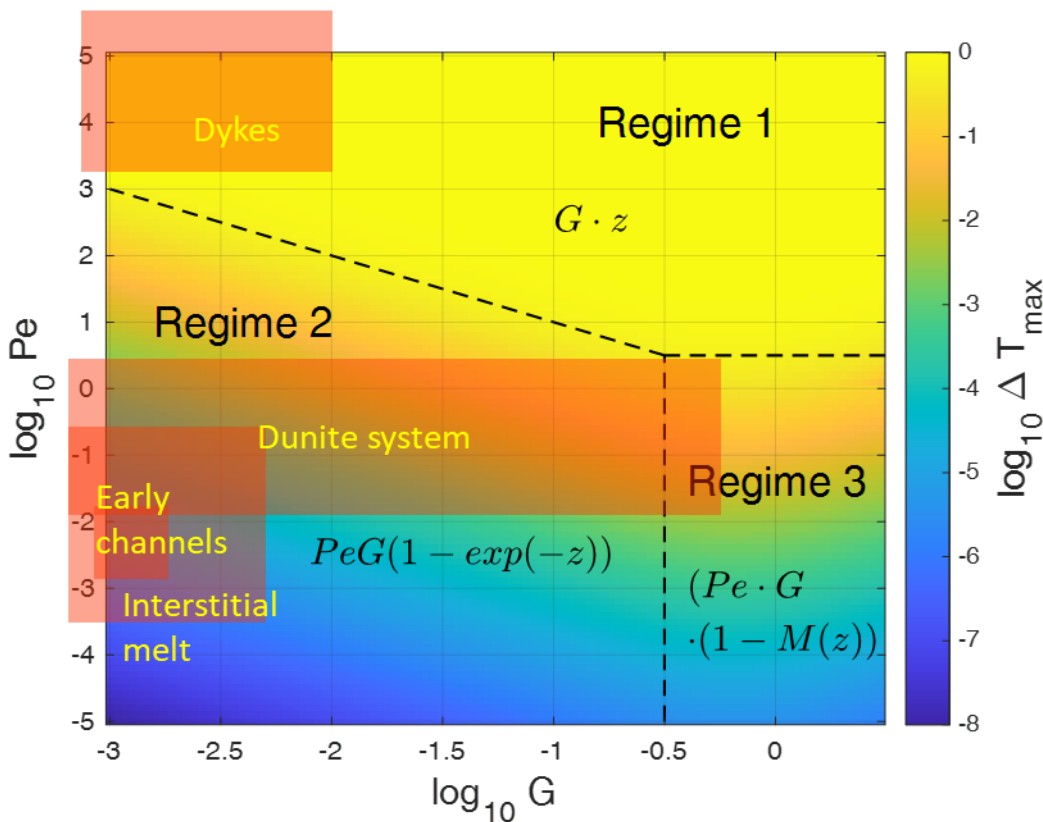

750

**Figure 4. Main regimes of the maximum fluid – solid temperature differences $\Delta T_{max}$ due to thermal non-equilibrium obtained by the analytical solution (Eq. 22) in the parameter space of the Peclet number $Pe$ and temperature gradient $G$. The asymptotic limits are indicated by the formulas, $M(z)$ is given by Eq. (27) with $\left(1 - M(G,z)\right)$ increasing non-linearly from about 0 to about 0.4 with increasing $z$ for $G$ in the range 0.3 to 3. Regime boundaries are shown as dashed lines. Typical parameter combinations for magmatic**
755 **settings such as interstitial melts or dykes are indicated by the orange rectangles which extend further to the left, well below $\log_{10} G$ of -3. Note that two slices through this field at $G = 0.1$ and at $Pe = 1$ have already been shown in Fig. 3.**

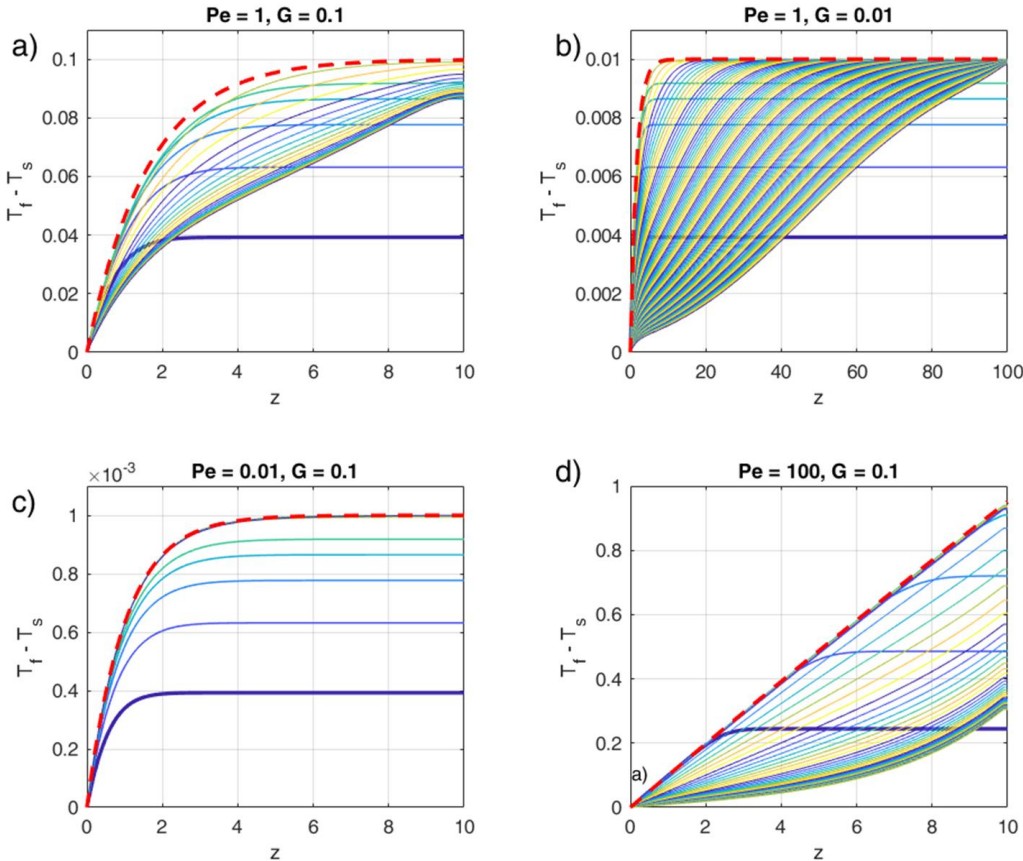

**Figure 5. Comparison of depth– and time- dependent numerical solutions with the time – independent analytical solutions for different parameters $Pe$ and $G$ as indicated in the sub-figure titles. In each panel the curves show $(T_f - T_s)$ - profiles for progressive times, the colors are cyclically varied with time from blue to yellow, starting with blue (bold curve). The bold red dashed curve shows the analytical solution Eq. (22), which represents a very good estimate of the depth-dependent temporal maximum of the temperature difference. In each panel the first 5 curves are plotted at time increments of 0.5 (0.025 for $Pe = 100$), the later curves with 5 (1 for $Pe = 100$). The total non-dimensional times of each panel are: 100 (500 for $G = 0.01$). Steady state is reached in the models with $G = 0.1$. The model with $G = 0.01$ would need $t = 10\ 000$ to reach steady state. The melt fraction was chosen as $\phi_0 = 0.1$.**

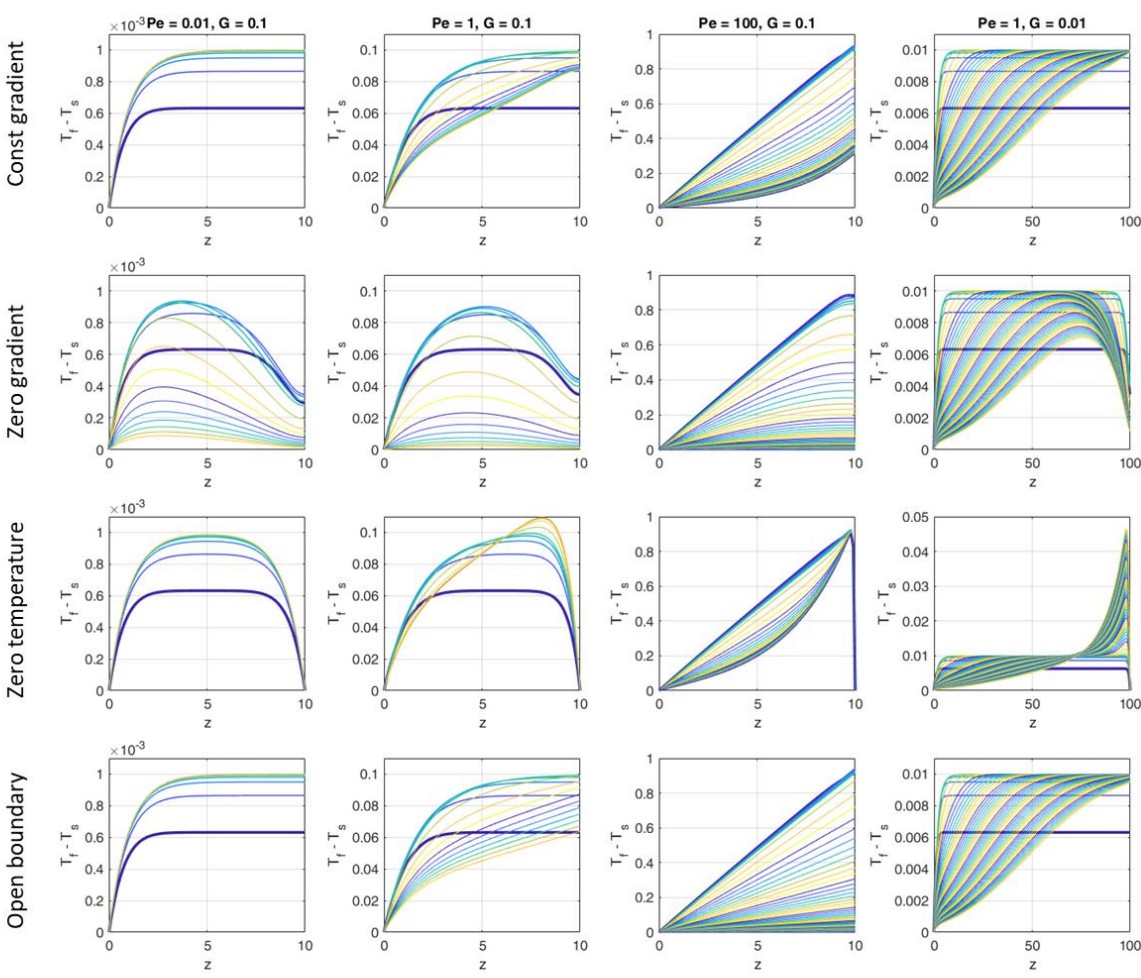

**Figure 6. Temporal evolution of vertical profiles of $(T_f - T_s)$ for models with different Peclet numbers and different initial temperature gradients $G$. In each panel the curves show $(T_f - T_s)$ - profiles for progressive times, the colors are cyclically varied with time from blue to yellow, starting with blue (bold curve). The first 5 curves of the $Pe < 100$ (respectively $Pe = 100$) models were taken with time increments of 1 (respectively 0.1), the later curves with 10 (respectively 1). The total time was 100 in all models with $G = 0.1$ and 500 in the models with $G = 0.01$. Steady state is reached in the models with $G = 0.1$. The models with $G = 0.01$ would need $t = 10\ 000$ to reach steady state. In each row the top boundary conditions is assumed as indicated at the left.**

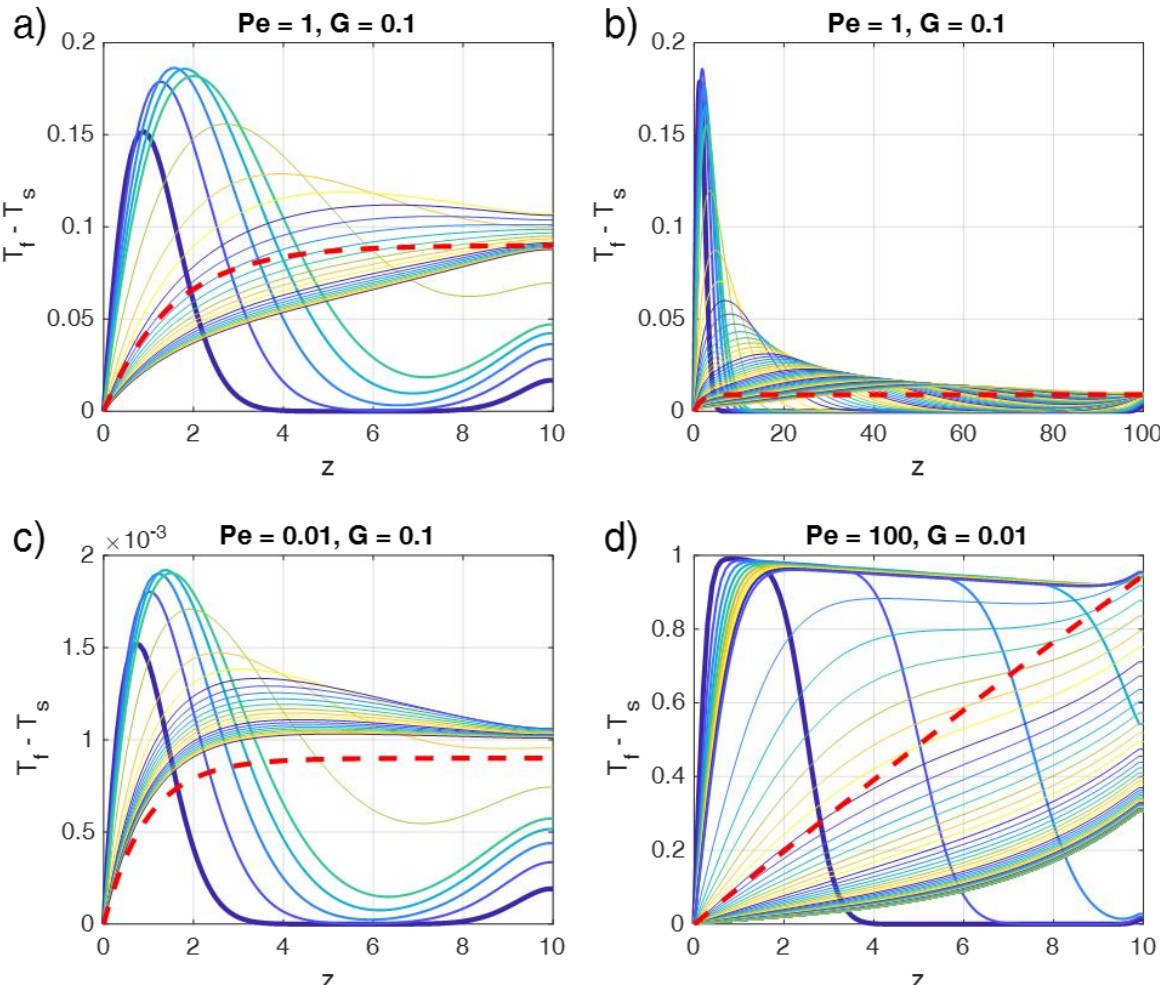

**Figure 7.** Time- and depth- dependent numerical solutions (thin curves) as in Figure 5 but for step-function initial conditions: $T_f = T_s = 1$ at $z = 0$ and $T_f = T_s = 0$ at $z > 0$ at $t = 0$. The bold dashed red curves are the time-independent analytical solutions as in Fig. 5. In each panel the curves show $(T_f - T_s)$ - profiles for progressive times, the colors are cyclically varied with time from blue to yellow, starting with blue (bold curve). In each panel the first 5 curves (and later curves, respectively) are plotted at time increments of a) 0.5 (5), b) 1 (10), c) 0.5 (5), and d) 0.025 (1). The total non-dimensional times of each panel are: 100 (500 for $G = 0.01$). Steady state is reached in the models with $G = 0.1$. The model with $G = 0.01$ would need $t = 10\,000$ to reach steady state. As porosity $\phi = 0.1$ is assumed.

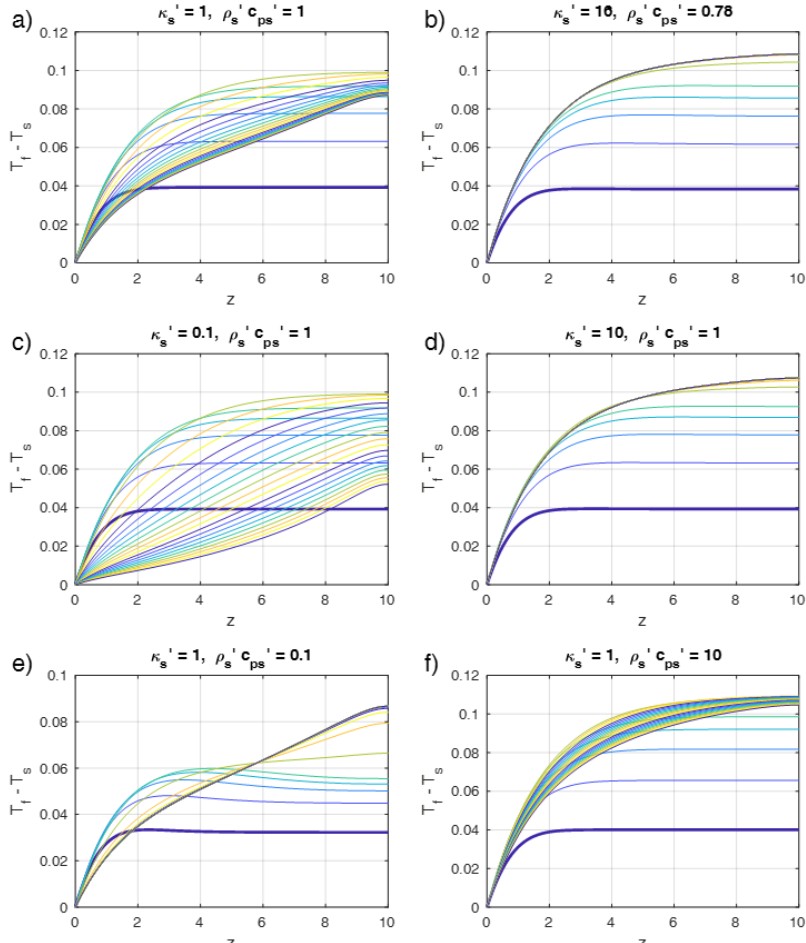

**Figure 8. Time- and depth- dependent profiles of the fluid – solid temperature differences as in Fig. 5. a) Reference models (as in Fig. 5a) with $Pe = 1$, $G = 0.1$, $\phi = 0.1$ and equal fluid to solid properties. b) to f) Profiles as in a) but with solid to fluid properties ratios as indicated in the titles of each panel, and $\lambda_{eff}' = 1$. The properties in b) are typical for water in sedimentary rocks. In each panel but b) the first 5 curves were taken with time increments of 0.5, the later curves with 5. In panel b) the first 5 curves were taken with time increments of 0.4875, the later curves with 4.875 . The total time was 100 in all models. Steady state is reached in all models.**

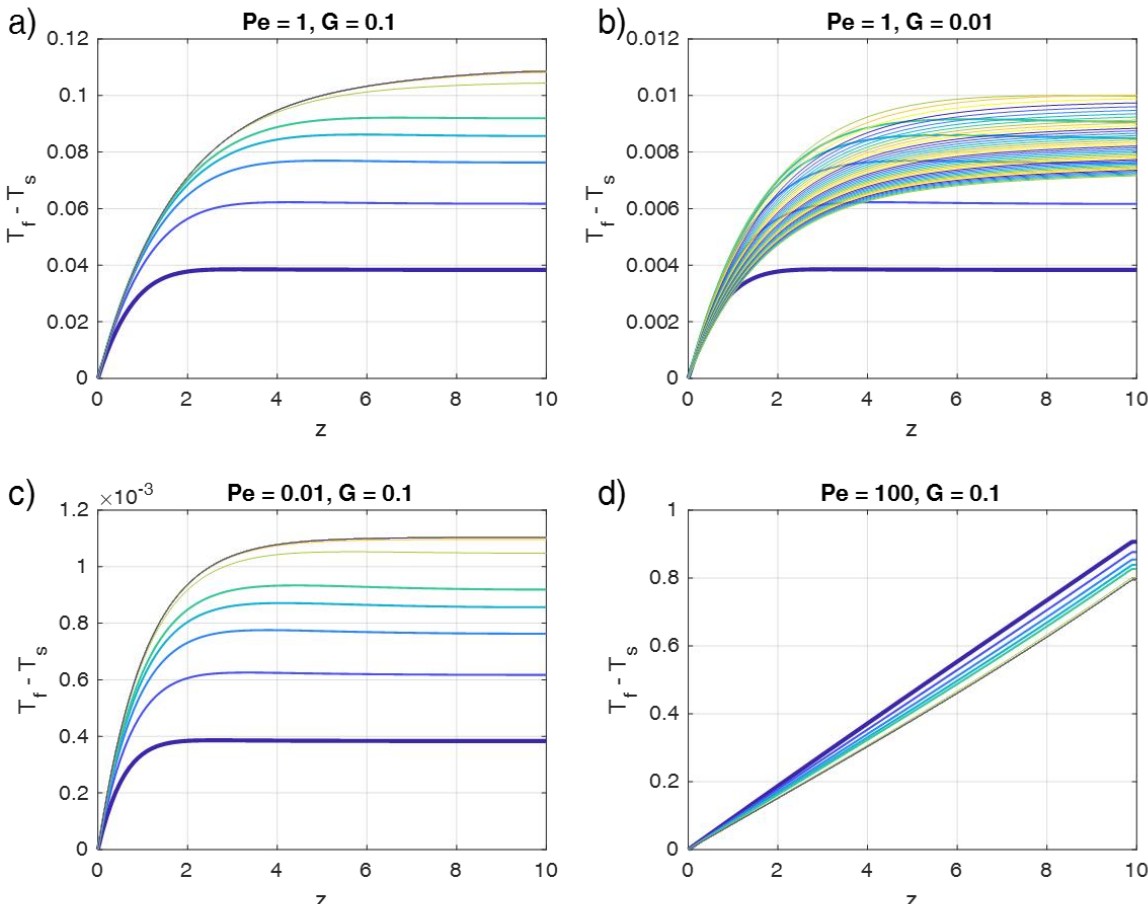

**Figure 9. Time- and depth- dependent profiles of the fluid – solid temperature differences as in Fig. 5, but for fluid to solid property ratios typical for water flowing through sedimentary rocks, i.e. $\rho_s' c_{p,s}' = 0.78$, $\kappa_s' = 16$ , $\lambda_{eff}' = 1$. $Pe$ and $G$ have been chosen as indicated in the sub-figure titles (as in Fig. 5) and $\phi = 0.1$ was assumed. In each panel the curves show $(T_f - T_s)$ - profiles for progressive times, the colors are cyclically varied with time from blue to yellow, starting with blue (bold curve). The first 5 curves were taken with time increments of 0.4875, the later curves with 4.875. The total time was 100 in all models with $G = 0.1$ and 200 in the models with $G = 0.01$. Steady state is reached in the models with $G = 0.1$. The model with $G = 0.01$ would need $t = 10\,000$ to reach steady state.**

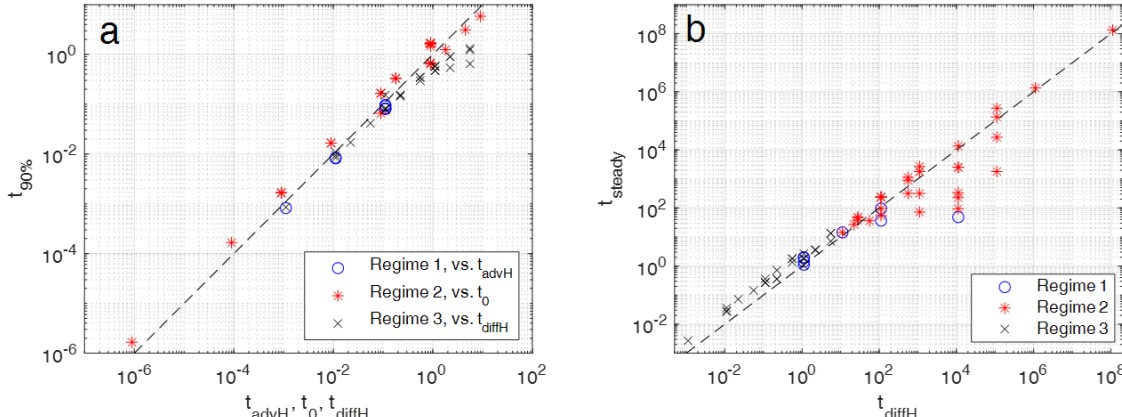

**Figure 10. For evaluating time scales the numerically determined times of models with various parameters *Pe* and *G* representing the three different regimes 1, 2 and 3 (different symbols) are plotted against characteristic scaling times. a) Times for reaching 90% of the maximum temperature difference $\Delta T_{max}$ are plotted against either the advective time scale $t_{advH}$ based on model height *H* for regime 1 models, or against the scaling time $t_0$ for regime 2 models, or against the diffusive time scale $t_{diffH}$ based on the model height *H*. b) times for reaching steady states are plotted against the characteristic diffusive time scales, $t_{diffH}$, based on model height *H* for all 3 regimes. Models close to the dashed line ($y = x$) are in best agreement with the characteristic times. In a) the Regime 2 times are taken dimensional by multiplying the observed times and the non-dimensional scaling time $t_0' = 1$ by some arbitrary dimensional times $t_0$.**