# Peer review of "Thermal non-equilibrium of porous flow in a resting matrix applicable to melt migration: a parametric study"

_Solid Earth, 2021_

## Referee Comment (RC1)

**Review of "Thermal non-equilibrium of porous flow in a resting matrix applicable to melt migration: a parametric study"**
**se-2021-149**

John Rudge

January 12, 2022

This manuscript presents a simple 1D model of the thermal equilibration of a fluid rising at a uniform velocity through a porous medium. Such models have been extensively studied in the engineering literature, but may be less familiar to the readers of Solid Earth. The authors apply their model to quantify the likelihood of thermal disequilibrium between melt and solid during different stages of melt ascent through the Earth's mantle: from diffuse porous flow, to channelised flow, to rapid ascent in dykes. Their main conclusions are perhaps not surprising: diffuse porous flow is in thermal equilibrium, dykes are not, and channels are somewhere in between. Indeed, this was also one of the main conclusions of a recent 2018 paper by one of the authors (Schmeling et al. 2018 GJI). The simple model presented here leads to some useful analytical expressions for quantifying thermal disequilibrium in geological systems. However, I think the presentation of the theory at the moment is somewhat overcomplicated, and a more streamlined revised manuscript could be produced.

**Major points**

1. The analysis of the thermal problem can be significantly simplified. At the moment, the results are presented in terms of four dimensionless numbers: Pe, $A$, $\phi$, and $H$. In fact just three dimensionless numbers are sufficient to describe the system: a modified Peclet number $\widetilde{Pe}$, a modified dimensionless domain height $\widetilde{H}$, and the porosity $\phi$.

   The authors in (8) choose to make lengths non-dimensional on $L_0$, which is a characteristic channel width of the pores. However, the equations are much simpler if one instead chooses to non-dimensionalise on a scale $L$ defined by

   $$L = \sqrt{\frac{\phi\delta}{S}} \tag{1}$$

   where $\delta$ is what the authors term the interfacial boundary layer thickness $dm$ and $S$ is the interfacial area density. $L$ is the natural lengthscale associated with thermal equilibration of melt pores. With time scaled by the diffusion time

   $$\tau = \frac{L^2}{\kappa} = \frac{\phi\delta}{\kappa S} \tag{2}$$

the dimensionless equations become

$$\frac{\partial T_f}{\partial t} + \widetilde{Pe}\frac{\partial T_f}{\partial z} = \frac{\partial^2 T_f}{\partial z^2} - (T_f - T_s) \tag{3}$$

$$\frac{\partial T_s}{\partial t} = \frac{\partial^2 T_s}{\partial z^2} + \frac{\phi}{1-\phi}(T_f - T_s) \tag{4}$$

where the modified Peclet number and dimensionless domain height are given by

$$\widetilde{Pe} = \frac{vL}{\kappa}, \qquad \widetilde{H} = \frac{H}{L} \tag{5}$$

where $H$ is the dimensional height of the domain and $v$ is the dimensional velocity of the upwelling melt. (3) and (4) are simpler than the authors (15) and (16) in containing just $\widetilde{Pe}$ and $\phi$ as dimensionless parameters, rather than the Pe, $A$, and $\phi$ that the authors have in (15) and (16). The authors' (15) and (16) relate to (3) and (4) by the transformation

$$\widetilde{Pe} = \frac{Pe}{A^{1/2}}, \qquad z'' = A^{1/2}z', \qquad t'' = t' \tag{6}$$

where the double primes refer to the non-dimensionalisation scheme used in this review, and single primes to that used by the authors.

Reducing the number of dimensionless parameters should allow the authors to greatly simplify their analysis. Instead of the talk of separate variation in the parameters Pe and $A$, and the Pe–$A$ parameter space, the behaviour can all be boiled down to variations in the single parameter $\widetilde{Pe}$. All mention of the separate parameter $A$ can be deleted.

2. It's not clear to me why the authors chose the boundary conditions they did, and I think these boundary conditions would benefit from a little explanation of how they relate to the Earth problems they're interested in. The authors describe their boundary conditions in lines 145-150. The conditions are a fixed temperature for both phases $T = \Delta T_0$ at the base of the column, and a fixed conductive heat flux $\frac{\partial T}{\partial z} = -\Delta T_0/H$ at the top. On line 148 the authors have missed a minus sign (temperature decreases with increasing height), and the authors are speaking too loosely by calling the top condition a constant flux condition. Because the melt is moving, there are two mechanisms of heat transfer out of the domain: the conductive (or diffusive) heat loss and the advective heat loss due to the moving fluid. The author's top boundary condition fixes just the conductive part. The authors should be clearer in talking about the advective and conductive heat losses, and provide some physical justification for their chosen boundary conditions. The same comment applies to line 331.

3. In the final discussion, I think it is worth writing some of the key results back in a dimensional form to make them more accessible. For example, in the cases where a constant temperature gradient in the solid is a good approximation, the key result is that a dimensional thermal disequilibrium of

$$T_f - T_s = -\frac{\phi(1-\phi)\delta vG}{\kappa S} \tag{7}$$

develops, where $G$ is the background temperature gradient. The characteristic time scale for the pore-scale thermal equilibration process is

$$\frac{\phi(1-\phi)\delta}{\kappa S} \tag{8}$$

Equations (7) and (8) are nice simple equations that can be used to get ball-park estimates of the expected thermal disequilibrium for geological problems.

**Minor points**

1. I think it's best to have a single letter variable, such as $\delta$, representing the boundary layer thickness, as the author's choice of $dm$ could be confused as a product of two other variables.

2. Line 232. The authors refer to "this surprisingly good agreement" between their numerics and the analytical approximation. I don't think they should be that surprised by the agreement! The analytical approximation is exploiting the large separation in time scales in the problem, with a fast timescale for pore-scale thermal equilibration, and a slow timescale for heat to be advected across the whole domain. The approximation could be formally justified by a multiple-scale analysis.

3. I don't understand the difference between the channels and dunite system marked in Figure 4. To me, the dunite system is a manifestation of channels.

4. Line 9. "by Darcy flow" – I'm not sure if that is a correct statement. Darcy's law is never used in the manuscript; a constant melt upwelling velocity is simply prescribed. Perhaps delete "by Darcy flow" here.

5. Line 56. As the authors note, these two-temperature equations have been very well studied in the engineering literature. It may be worth also citing some of the work of Kuznetsov (e.g. Kuznetsov 1994 Int J. Heat Mass Transfer 37 5050-5055) and Spiga and Spiga 1981 Int J. Heat Mass Transfer 24 355-364.

6. I think the presentation of the heat equations can be streamlined a little. I'd start by writing (4) and (5) and not write (1), (2), or (3). It may be worth noting that these energy equations are for constant pressure.

7. Figure 2. It would be good to have some indication as to what times the different lines in panel a correspond to. Perhaps add a legend?

**Typos**

1. Line 121. Besides is not right here.

2. Line 273. L'Hôpital not L'Hospital.

3. Line 315. onm

4. Line 373. affected not effected

5. There are a number of places where the English phrasing is a little odd, and the manuscript could be benefit from some additional proof-reading.

---

## Referee Comment (RC2)

**Thermal non-equilibrium of porous flow in a resting matrix applicable to melt migration: a parametric study**

Cian Wilson

January 2022

The manuscript describes thermal non-equilibrium in two-phase systems using a simplified one-dimensional model where the solid is stationary, fluid velocity is constant and prescribed, and material parameters are constant. The authors take the additional step of seeking analytical solutions near the peak of disequilibrium by imposing a thermal gradient in the solid phase and assuming a steady state. This allows them to examine the limits of the equations and examine the behavior in several important regimes. This is done quite rigorously with significant discussion of several scenarios.

Overall the manuscript is rigorous and compelling but I agree with John Rudge's review and think it could be significantly streamlined. This would make the impressive body of work much more digestible and useful to the wider community.

**Scaling**

As John pointed out the number of parameters could be reduced with a redefinition of the length-scale $L_0$. This would have a huge impact on all the subsequent sections, making the conclusions much more tractable. I would encourage the authors to take a small extra step and redefine the length-scale such that:

$$L_0^2 = \frac{\delta}{S}, \tag{1}$$

where I have adopted John's notation and used $\delta = dm$ and have dropped the $\varphi$ dependence. The equations then become:

$$\frac{\partial T_f}{\partial t} + Pe\boldsymbol{v} \cdot \nabla T_f = \nabla^2 T_f - \frac{1}{\varphi}(T_f - T_s) \tag{2}$$

$$\frac{\partial T_s}{\partial t} = \nabla^2 T_s + \frac{1}{1 - \varphi}(T_f - T_s) \tag{3}$$

$$\tag{4}$$

with controlling parameters:

$$Pe = \frac{L_0 v_0}{\kappa_0} = \sqrt{\frac{\delta}{S}} \frac{v_0}{\kappa_0} \quad \text{and} \quad \varphi \tag{5}$$

This has the small additional appeal of removing the $\varphi$ dependence from $Pe$ (or $A$) and having independent controlling parameters.

**Boundary Conditions**

The top boundary condition is frequently referred to as a constant flux condition but it appears to only set the diffusive part of the condition and ignores any discussion of the advective component. This has large

implications for the behavior at the top of the domain, where most of the discussion is focused because it is where the largest (or most persistent) non-equilibrium conditions are seen (at least in the case with a linear initial condition). The persistence of the disequilibrium is a product of the boundary conditions however, with the solid temperature being prevented from further equilibrating with the fluid by its flux being fixed. I was a little confused by the physical motivation of this boundary condition where fluid is (presumably) allowed to escape but the solid temperature gradient is fixed and would like to have it discussed or justified more at its introduction. The authors do later vary the boundary condition in the discussion and supplement but again don't offer physical motivations for the options and again don't discuss the advective flux component.

**Analytical Justification**

In section 4.1 the authors simplify their model to seek the temperature difference around a fixed solid temperature profile. The description of this simplification could be improved. For a start I believe the solid temperature gradient has a sign error on line 210 $\partial T_s/\partial z = -\Delta T/H$ (the subsequent equations are correct) and, as far as I could see, $\Delta T$ has not been defined at this point. $\Delta T_0$ was previously used for the temperature drop across the domain (and $\Delta T_{max}$ for the peak non-equilibrium temperature difference). Beyond that more clearly stating that this fixes the solid temperature gradient and seeks a solution in this state would make reading this section easier (the authors do try to clarify that the steady state behavior they are seeking isn't the same as that in the "full" model).

**Discussion**

The discussion section rigorously describes a number of different scenarios and variations on the previous results. It covers an impressive range of topics though section 5.1.5 seems redundant and could be removed. It would be hugely enhanced by reducing the complexity and only considering a two (with presumably only one important) parameter model. Currently a full understanding of the section depends on several figures that are only available in the supplementary material, which are not particularly well captioned. If it were possible to improve the labeling and move them or a subset of them to the main text that would enhance the discussion significantly. In particular, though the color coded time evolution works well when lines are close together, it fails to clearly show the order of the solutions once significant jumps occur between them. It would also be good to know what (non-dimensional) time-ranges are being plotted in all figures.

**Minor Points**

There are quite a few small typos throughout the paper (such as extra or missing spaces) that should be corrected. Below I just list a few things that caught my eye:

- line 72: "allowing easily to decide" $\rightarrow$ "that allow the easy determination of"?

- start of section 2.1: a constant porosity and fluid velocity are already stated here but aren't used in the following derivation, instead being repeated at the end of the section (where they are used)

- line 86 and throughout where appropriate: "resumes to" $\rightarrow$ "becomes"

- equation 9a: $\kappa_0$ is introduced but isn't defined until equation 11 and even then is related to $\kappa_f$ and $\kappa_s$, which also haven't been introduced before. I think a statement that $\lambda_f = \lambda_s = \lambda_{eff}$ might be necessary first.

- line 134: "is to emphasized" $\rightarrow$ "is to be emphasized"

- start of section 2.3: the domain height, $H$ is introduced before it is used, perhaps consider adding to the first sentence: "are solved in a 1D domain of height $H$"

- line 148: sign error in "flux" boundary condition

- line 152: extra space in "R2018b )"

- lines 169-171: the sentence "As the fluid temperature increases" could do with rewriting for clarity.

- line 177: "boundary conditions are applied" → "boundary conditions at $z = H$ are applied"

- line 210: sign error in solid temperature gradient and $\Delta T$ introduced without definition

- line 224: missing space between "for" and "$(T_f - T_s)$"

- line 231: "For 91% of the models" - does this include those models with a different initial condition or those with varying material properties?

- line 245: I think "complex" could be confusing here, maybe try "complicated"

- line 273: "L'Hospital" → "L'Hôpital"

- line 287: extra space after "dykes"

- line 315: "onm" → "on"

- line 340: "As initial condition" → "As an initial condition"

- section 5.1.4: redefining $A$ for this section is potentially a little confusing. Hopefully $A$ is no longer necessary anyway but, if not, perhaps in earlier sections the constant version could be $A_0$ so that the variable wouldn't have to be re-used.

- section 5.1.5: this seems redundant and could be removed

- line 428: it might be nice here to clarify that this is the steady state of the full model, not the analytic model

- supplementary material line 23: this looks like it's the solution of Eq. (19) from the main text not the homogeneous Eq. (S2)

- line 46: "into Taylor series" → "into a Taylor series"

- Figure S4 caption: spacing and punctuation errors

---

## Author Comment (AC1)

**Comments by authors to the reviews of "Thermal non-equilibrium of porous flow in a resting matrix applicable to melt migration: a parametric study"**

Harro Schmeling

March, 4th, 2022

We thank both reviewers for their thoughtful and constructive reviews. We extensively rewrote the paper according to the comments by the reviewers. In the following we react point by point to the comments which are given in italics.

**Review Rudge**

1.) ... Reducing the number of dimensionless parameters should allow the authors to greatly simplify their analysis. Instead of the talk of separate variation in the parameters Pe and A, and the Pe-A parameter space, the behaviour can all be boiled down to variations in the single parameter  $\widetilde{Pe}$ . All mention of the separate parameter A can be deleted.

This is an excellent point! Indeed, we used different lengths for the scaling length and the diffusion length used to define the scaling time. We adopt Rudge's suggestion and use the diffusion length not only for the scaling time as in the previous version but also for the scaling length *L*. Indeed, we found out that using a modification of Rudge's scaling length *L* leads to a more elegant non-dimensionalization:

$$L = \sqrt{\frac{\phi_0(1-\phi_0)\delta}{s}}$$

where  $\phi_0$  is the melt fraction,  $\delta$  is the interfacial boundary layer thickness and *S* is the interfacial area density. This scaling length leads to more symmetric heat equations for solid and fluid (new eq. 14, 15), respectively, and to an equation for the fluid to solid temperature difference independent of  $\phi_0$  (new equation 20)! Thus, for the non-equilibrium temperature difference the problem depends only on two rather than 4 non-dimensional parameters, the Peclet number *Pe* and the non-dimensional initial temperature gradient *G*. The previously used additional parameters *A* and  $\phi_0$  drop out in equation 20, and *A* drops out of the equations for solid and fluid temperatures (new equ. 4 and 5). We now emphasize that this scaling diffusion length scales with the geometric mean of the pore dimension and the interfacial boundary layer thickness. With the new scaling the heat exchange parameter *A* drops out. However, our calculated numerical models are not obsolete, because the previous variation of *A* and  $\phi_0$ maps into a variation of *Pe* and *H*, where *H* is the thickness of the model. Indeed, in the original paper we had 4 non-dimensional parameters, *Pe*, *A*,  $\phi$  and *H* and focused on the *Pe* – *A* space. With the new scaling, only three of these four parameters are independent, and for the temperature difference only two are independent. Following his suggestion we dropped *A* everywhere and focus on the *Pe* – *G* (=  $\frac{1}{H}$ ) parameter space. The three different regimes seen in the previous version show up again as three modified regimes. Where necessary we have rerun most models with the new scaling. We dropped all models with  $A \neq 1$  but included some important cases with various *H*. Only for section 5.2 (Time scales) we still use the results of the old models mapped to new Peclet numbers and thicknesses *H* but added a number of new parameter combinations.

2. It's not clear to me why the authors chose the boundary conditions they did, and I think these boundary conditions would benefit from a little explanation of how they relate to the Earth problems they're interested in. The authors describe their boundary conditions in lines 145-150. The conditions are a fixed temperature for both phases  $T = \Delta T_0$  at the base of the column, and a fixed conductive heat  $flux \frac{\partial T}{\partial z} = -\Delta T_0/H$  at the top. On line 148 the authors have missed a minus sign (temperature decreases with increasing height), and the authors are speaking too loosely by calling the top condition a constant ux condition. Because the melt is moving, there are two mechanisms of heat transfer out of the domain: the conductive (or diffusive) heat loss and the advective heat loss due to the moving fluid. The author's top boundary condition fixes just the conductive part. The authors should be clearer in talking about the advective and conductive heat losses, and provide some physical justification for their chosen boundary conditions. The same comment applies to line 331.

Good point. We now introduce the boundary condition at the top more rigorously, and address the advective flux. We justify the chosen boundary condition being appropriate except for late stages. In chapter 5.1.2 we extend the discussion about the limitations, and test three alternative boundary conditions, including an open boundary. We also justify that the desired temperature differences are well given by the values at the boundary despite the discussed limitations.

3. In the final discussion, I think it is worth writing some of the key results back in a dimensional form to make them more accessible. For example, in the cases where a constant temperature gradient in the solid is a good approximation, the key result is that a dimensional thermal disequilibrium of

 $T_f - T_s = -\frac{\phi(1-\phi)\delta vG}{\kappa S} \tag{7}$

develops, where G is the background temperature gradient. The characteristic time scale for the porescale thermal equilibration process is

 $\frac{\phi(1-\phi)\delta}{\kappa S}$

(8)

Equations (7) and (8) are nice simple equations that can be used to get ball-park estimates of the expected thermal disequilibrium for geological problems.

Nice suggestion. We now include an appropriate section at the end of the application section.

Minor points

1. I think it's best to have a single letter variable, such as  $\delta$ , representing the boundary layer thickness, as the author's choice of dm could be confused as a product of two other variables.

Agreed and changed

2. Line 232. The authors refer to "this surprisingly good agreement" between their numerics and the analytical approximation. I don't think they should be that surprised by the agreement! The analytical approximation is exploiting the large separation in time scales in the problem, with a fast timescale for pore-scale thermal equilibration, and a slow timescale for heat to be advected across the whole domain. The approximation could be formally justifed by a multiple-scale analysis.

We replaced "surprisingly" by "very good". But we cannot confirm that the large separation of time scales is the reason. Some tests with similar or reversed timescales (Pe = 1, H = 1 or Pe = 0.1 and H=1, respectively) give similarly very good agreement (not included). We don't carry out a multiple-scale analysis.

3. I don't understand the difference between the channels and dunite system marked in Figure 4. To me, the dunite system is a manifestation of channels.

O.k., we distinguish between the onset of channeling and late stages. This is done in the text and we now write "early channels" in Fig 4.

4. Line 9. "by Darcy flow" - 1'm not sure if that is a correct statement. Darcy's law is never used in the manuscript; a constant melt upwelling velocity is simply prescribed. Perhaps delete "by Darcy ow" here.

Agreed. Replaced by "porous flow"

5. Line 56. As the authors note, these two-temperature equations have been very well studied in the engineering literature. It may be worth also citing some of the work of Kuznetsov (e.g. Kuznetsov 1994 Int J. Heat Mass Transfer 37 5050-5055) and Spiga and Spiga 1981 Int J. Heat Mass Transfer 24 355-364.

Thanks for the suggestions, included now.

6. I think the presentation of the heat equations can be streamlined a little. I'd start by writing (4) and (5) and not write (1), (2), or (3). It may be worth noting that these energy equations are for constant pressure.

We prefer to start with (1) - (3) because then it is clear that the equations still allow for variable material properties (even  $c_p$  !). We mention the variable properties now (not  $c_p$  explicitly) and mention the constant pressure.

7. Figure 2. It would be good to have some indication as to what times the different lines in panel a correspond to. Perhaps add a legend?

We have redrawn the figure completely with the new scaling and added the times

Typos

Typos corrected and English improved.

**Review Wilson**

**1.) Scaling**

As John pointed out the number of parameters could be reduced with a redefinition of the length-scale LO. This would have a huge impact on all the subsequent sections, making the conclusions much more tractable. I would encourage the authors to take a small extra step and redefine the length-scale such that:

$$L_0^2 = \frac{\delta}{s} \tag{1}$$

where I have adopted John's notation and used  $\delta$  = dm and have dropped the  $\varphi$  dependence. The equations then become:

$$\frac{\partial T_f}{\partial t} + Pev \cdot \nabla T_f = \nabla^2 T_f - \frac{1}{\varphi} (T_f - T_s)$$

$$\frac{\partial T_s}{\partial t} = \nabla^2 s + \frac{1}{1-\varphi} (T_f - T_s)$$
(2)
with controlling parameters:

with controlling parameters:

$$Pe = \frac{L_0 v_0}{\kappa_0} = \sqrt{\frac{\delta}{S} \frac{v_0}{\kappa_0}} \qquad \text{and} \qquad \varphi \tag{5}$$

This has the small additional appeal of removing the  $\varphi$ -dependence from Pe (or A) and having independent controlling parameters.

This is an alternative scaling, but still the length scale is implicitly dependent on  $\varphi$  because *S* depends on  $\varphi$ . As we want to use melt channels, tubes or melt films as typical melt geometries one can write  $S = c \varphi/d_f$  with  $d_f$  as melt pore dimension and c = 2 or 4 for channels, films or tubes. Motivated by your and Rudge's suggestions about scaling we came up with another more elegant alternative which makes the two heat equations for solid and fluid more symmetric with respect to each other, and which leads the porosity to drop out the equation for the temperature difference. See also our reply to comment 1 by Rudge. With this scaling the length scale is essentially equal to the geometric mean of  $\delta$  and  $d_f$  or  $d_s$  which Rudge calls a natural length scale for thermal melt equilibration. We make this point in the revised manuscript.

**2.) Boundary Conditions**

The top boundary condition is frequently referred to as a constant flux condition but it appears to only set the diffusive part of the condition and ignores any discussion of the advective component. This has large implications for the behavior at the top of the domain, where most of the discussion is focused because it is where the largest (or most persistent) non-equilibrium conditions are seen (at least in the case with a linear initial condition). The persistence of the disequilibrium is a product of the boundary conditions however, with the solid temperature being prevented from further equilibrating with the fluid by its flux being fixed. I was a little confused by the physical motivation of this boundary condition where fluid is (presumably) allowed to escape but the solid temperature gradient is fixed and would like to have it discussed or justified more at its introduction. The authors do later vary the boundary condition in the discussion and supplement but again don't offer physical motivations for the options and again don't discuss the advective flux component.

Good point. We now introduce the boundary condition at the top more rigorously, and address the advective flux at its introduction. There we justify the chosen boundary condition being appropriate except for late stages. In chapter 5.1.2 we extend the discussion about the limitations, and test three alternative boundary conditions, including an open boundary. We also justify that the desired temperature differences can safely be picked by the values at the boundary despite the discussed limitations.

**3.) Analytical Justification**

In section 4.1 the authors simplify their model to seek the temperature difference around a fixed solid temperature profile. The description of this simplification could be improved. For a start I believe the solid temperature gradient has a sign error on line 210  $\partial T_s / \partial z = -\Delta T_0 / H$  (the subsequent equations are correct) and, as far as I could see,  $\Delta T$  has not been defined at this point.  $\Delta T_0$  was previously used for the temperature drop across the domain (and  $\Delta T_{max}$  for the peak non-equilibrium temperature difference). Beyond that more clearly stating that this fixes the solid temperature gradient and seeks a solution in this state would make reading this section easier (the authors do try to clarify that the steady state behavior they are seeking isn't the same as that in the "full" model).

Thanks! We corrected the sign error and dropped  $\Delta T$  which was not defined, indeed. We don't mix up dimensional and non-dimensional quantities in that section anymore. At two places in section 4.1 we added "fixed" to the temperature gradient in the solid.

**4.) Discussion**

The discussion section rigorously describes a number of different scenarios and variations on the previous results. It covers an impressive range of topics though section 5.1.5 seems redundant and could be removed. It would be hugely enhanced by reducing the complexity and only considering a two (with presumably only one important) parameter model. Currently a full understanding of the section depends on several figures that are only available in the supplementary material, which are not particularly well captioned. If it were possible to improve the labeling and move them or a subset of them to the main text that would enhance the discussion significantly. In particular, though the color coded time evolution works well when lines are close together, it fails to clearly show the order of the solutions once significant jumps occur between them. It would also be good to know what (non-dimensional) time-ranges are being plotted in all figures.

Agreed: we dropped section 5.1.5. As with the new scaling the quantity *A* disappeared , sections 4.3.1, 4.3.2, 5.1.4 could be dropped. From the supplementary material sections sections 1.4, 2.1, 2.2, and 5 have been dropped. The remaining figure captions are improved. The Figures have been redrawn with better coding the times of the curves (starting curve is bold now), and all times are mentioned in the captions now. We moved all figures but one from the supplementary material to the paper.

5.) Minor Points

There are quite a few small typos throughout the paper (such as extra or missing spaces) that should be corrected. Below I just list a few things that caught my eye:

. line 72: "allowing easily to decide"  $\rightarrow$  "that allow the easy determination of"? done

. start of section 2.1: a constant porosity and fluid velocity are already stated here but aren't used in the following derivation, instead being repeated at the end of the section (where they are used) o.k. done

. line 86 and throughout where appropriate: "resumes to" ightarrow "becomes" done

. equation 9a:  $\kappa_0$  is introduced but isn't defined until equation 11 and even then is related to  $\kappa_f$  and

 $\kappa_s$ , which also haven't been introduced before. I think a statement that  $\lambda_f = \lambda_s = \lambda_{eff}$  might be

necessary first. We refer to Table 1 and explicitly state equal thermal properties done

. line 134: "is to emphasized"  $\rightarrow$  "is to be emphasized" done

. start of section 2.3: the domain height, H is introduced before it is used, perhaps consider adding to the first sentence: "are solved in a 1D domain of height H" done

. line 148: sign error in "flux" boundary condition done

. line 152: extra space in "R2018b )" done

. lines 169-171: the sentence "As the fluid temperature increases" could do with rewriting for clarity.done

. line 177: "boundary conditions are applied"  $\rightarrow$  "boundary conditions at z = H are applied" done

. line 210: sign error in solid temperature gradient and  $\Delta T$  introduced without definition done

. line 224: missing space between "for" and " $(T_f - T_s)$ "done

. *line 231: "For 91% of the models" - does this include those models with a different initial condition or those with varying material properties?* no, for these no analytical solution exists. Reformulated. done

. line 245: I think "complex" could be confusing here, maybe try "complicated" done

. line 273: "L'Hospital" → "L'H^opital" done

. line 287: extra space after "dykes" done

. *line 315: "onm*" → "on"done

. line 340: "As initial condition"  $\rightarrow$  "As an initial condition" done

. section 5.1.4: redefining A for this section is potentially a little confusing. Hopefully A is no longer necessary anyway but, if not, perhaps in earlier sections the constant version could be  $A_0$  so that the variable wouldn't have to be re-used. obsolete

. section 5.1.5: this seems redundant and could be removed done

. *line 428: it might be nice here to clarify that this is the steady state of the full model, not the analytic model.* The analytic model is independent of time. Added "numerical model" done

. supplementary material line 23: this looks like it's the solution of Eq. (19) from the main text not the homogeneous Eq. (S2) done

. line 46: "into Taylor series"  $\rightarrow$  "into a Taylor series" obsolete, done

. Figure S4 caption: spacing and punctuation errors done

---

## Referee Report (RR1)

A review of "Thermal non-equilibrium of a porous flow in a resting matrix applicable to melt migration: a parameteric study", by Chevalier and Schmeling

In this paper, the authors present a simple 1D model of heat transport in a system consisting of a fluid moving with a constant velocity that is in contact with a solid. There is initially a constant temperature gradient decreasing from the bottom of the column to the top. Fluid is introduced at the bottom at a temperature equal to the bottom temperature. As heat is advected upward with the fluid, a temperature difference between the fluid and solid can result. The authors show that there are three distinct times, one in which the degree of disequilibrium is increasing, one in which it is steady, and one where it is falling to a steady state. The authors derive a simplified analytical model to predict the temperature difference between the liquid and solid in their model and show that it agrees well with their more complex, time dependent model.

The paper has been reviewed twice already by John Rudge and Cian Wilson who both suggested alternative non-dimensionalization of the equations. The authors have revised their paper based on the comments of the previous reviews and have changed the non-dimensionalization. I believe that the paper is new and interesting and that the authors have responded adequately to the two previous reviews. I have read the paper and made detailed comments on an annotated copy of the manuscript that I a returning.

I have two substantive comments regarding the content of the paper. The first is that I am surprised that the authors do not show any analyses of the heat flow in their model. If nothing else, the authors could show the overall energy balance in their model by showing the incoming and outgoing advected and conducted heat flows at the top and bottom as well as the change in internal energy in the solid and fluid. This would serve as a check on energy conservation in their model but analyses could also be done on smaller scales looking at heat fluxes between the solid and fluid as well as heat fluxes due to advection and conduction. This would serve to help explain the cause of nonequilibrium in their models and I think would add significantly to a reader's understanding.

My second substantive point regards the length scale in the model. As the authors show in equation 9a, the length scale is (phi delta ds/cs)^0.5 where phi is porosity, delta is the thermal boundary layer thickness, ds is the grain size and cs is a constant of order 1. Since the thermal boundary layer thickness will be small compared with the size of the system for large Peclet numbers and the grain size is very small compare with the size of the system, L will also be very small compared with the size of the system. In estimating G for a mid-ocean ridge the authors get values of 10^-6-0.6 for various settings. Yet, when investigating their models (figures 2-9) the authors only show values of G as low as 0.01. Values of z can only be as large as 1/G. I think that this issue with the length scale should be discussed. Also, as shown in figure 4, the degree of disequilibrium is proportional to G in all cases. In Regime 1, DeltaTmax is G\*z where G is small, the maximum value of z will be large. However, in Regimes 2 and 3, when G is small, unless Pe is very large, the degree of disequilibrium is likely to be very small based on the scaling laws in these two regimes since (1-exp(-z)) and M(z) are both of order 1,

The authors have already performed substantial revisions to their paper. I am not requiring that the above issues be addressed but if they could, I believe that the paper would be significantly improved.

[revised manuscript text omitted]

---

## Referee Report (RR2)

**Thermal non-equilibrium of porous flow in a resting matrix applicable to melt migration: a parametric study**

Cian Wilson

May 2022

The authors have made substantial changes to their original submission and the resulting manuscript now presents their work in a much more useful manner. Thank you. I'm still a little confused as to the applicability of the choices of initial and boundary conditions to the Earth but understand that these were likely made to make the analytical solutions tractable and the authors do a good job investigating what effects these decisions make using their numerical models. I only have minor comments and questions.

**Non-dimensional Numbers**

The non-dimensional parameters controlling the system has now been reduced to $Pe$, $\varphi$ and $G$ and only $Pe$ and $G$ in the analytical case. However, models are still presented in terms of $H$. Even though this is just the reciprocal of $G$ it would be great to use a single labeling system. $G$ and $H$ are also listed in table 1 as dimensional numbers but, I think, only discussed in their non-dimensional form (except maybe when discussing the scale length $L$). Even though primes are explicitly dropped this might be a little confusing.

**Initial & Boundary Conditions**

The dependence on $G'$ (or $H'$) really emphasized to me how dependent these solutions are on the initial and boundary conditions selected (I guess most problems are!). Though some discussion has been added to section 2.3 about the need to specify boundary conditions in general, this isn't really the kind of physical justification for this specific problem that I was hoping for. This model imagines an initially stagnant fluid in thermal equilibrium with a solid with a (Cartesian) conductive temperature profile. The fluid then moves and while the upper boundary allows advective outflow of heat in the fluid it maintains a fixed thermal gradient in both the solid and fluid while doing so. (All of this is much more clearly stated in the updated manuscript, thank you.) The analytical model then seeks the maximum thermal disequilibrium, which occurs in the early stages of the evolution of the model and at the top of the domain. I understand that these decisions were made to make the analytical solution possible but if there is a physical scenario in which this is likely then it would be great to discuss it in section 2.3.

The authors demonstrate numerically the effects that different initial and boundary conditions might have in section 5.1.2. This only seems to consider applying the same conditions to both the solid and fluid temperature. Have the authors considered applying different conditions to the two fields? This may make a more physical scenario possible. For example the fluid temperature could use the open boundary condition, mimicking fluid escape, while the solid could use a Robin condition relating the gradient at the top boundary to an imagined crustal conductive profile above the domain depending on the temperature at the top boundary, e.g. $\frac{\partial T}{\partial z} = \frac{T_{surf} - T}{h_{crust}}$. Admittedly this would introduce an extra parameter for the "crust" thickness $h_{crust}$ and the surface temperature $T_{surf}$.

**Time Evolution & Steady State**

I appreciated the authors' efforts to clarify the time evolution in the figures by giving some time increments and distinguishing the lines a little. In many cases the total time was listed for all models. Could the authors confirm that in the time evolution figures (e.g. Figure 6) the steady state solution (as opposed to just the solution at an arbitrary final time) is included in all subplots?

**Even More Minor Points**

There are still quite a few small typos throughout the paper that should be corrected in the editing phase. Below are a few things that caught my eye:

- line 31: "On small scale" → "On a small scale"

- line 67: "or more general" → "or more generally"

- line 113: Perhaps introduce a new paragraph after "Primed quantities are non-dimensional."?

- line 179: is the second statement about resolution only necessary for a few of the numerical cases presented in figure 3?

- equations 21, 22, 23: $G$ appears inconsistently italicized in these three equations (see line 331 also)

- equations 32 & 33: the non-dimensionalized prime is placed inconsistently throughout the variables in these equations, sometimes as a subscript and also as a superscript

- table 1, last line: "reference. respectively" → "reference, respectively"?

- figure 3: could figure S1 be incorporated into figure 3a to avoid having to reference a supplemental figure? Also, while figure 3 is necessary to show the numerical results could it reference figure 4 (and vice versa) as it (I believe) represents slices through the latter?

---

## Author Response (AR2)

**Comments by authors to the reviews of "Thermal non-equilibrium of porous flow in a resting matrix applicable to melt migration: a parametric study"**

Harro Schmeling

May, 27, 2022

We thank both reviewers for their thoughtful and constructive reviews. We rewrote the paper according to the comments by the reviewers. In the following we react point by point to the comments which are given in italics.

**Review Butler**

1.) I have two substantive comments regarding the content of the paper. The first is that I am surprised that the authors do not show any analyses of the heat flow in their model. If nothing else, the authors could show the overall energy balance in their model by showing the incoming and outgoing advected and conducted heat flows at the top and bottom as well as the change in internal energy in the solid and fluid. This would serve as a check on energy conservation in their model but analyses could also be done on smaller scales looking at heat fluxes between the solid and fluid as well as heat fluxes due to advection and conduction. This would serve to help explain the cause of nonequilibrium in their models and I think would add significantly to a reader's understanding.

We now include two extra panels in Fig. 2 showing the three contributions of the heat fluxes demonstrating the importance of advective heat flux by the fluid. Based on these new figures we add a short statement about the neglect of conductive terms which has been done in earlier papers. I am not so sure whether we can learn much more from more extensive heat flow analyses, so this should be sufficient. As for the heat balance, yes, we tested it and make a short statement in the Numerical scheme section. Fig. 1 below shows the global heat balance which is of error order 1 (due to upwind). However, we think it is not necessary to include it in the paper.

Figure 1. Left: Incoming heat flux (conductive + advective) at bottom, outgoing heat flux (conductive + advective) at top, and change rate of internal heat (Del-q) as a function of time of model 1. The different resolutions plot essentially on the same curves. Right: Error of the relative heat balance of these contributions with different resolutions corresponding to 101, 201, and 401 grid points.

2.) My second substantive point regards the length scale in the model. As the authors show in equation 9a, the length scale is (phi delta ds/cs)^0.5 where phi is porosity, delta is the thermal boundary layer thickness, ds is the grain size and cs is a constant of order 1. Since the thermal boundary layer thickness will be small compared with the size of the system for large Peclet numbers and the grain size is very small compare with the size of the system, L will also be very small compared with the size of the system. In estimating G for a mid-ocean ridge the authors get values of 10^-6-0.6 for various settings. Yet, when investigating their models (figures 2-9) the authors only show values of G as low as 0.01. Values of z can only be as large as 1/G. I think that this issue with the length scale should be discussed.

Perhaps this concern arises from the understanding that " $d_s$  is the grain size". In our model we introduce and understand  $d_s$  as the average distance between melt filled pores or channels which can be considerably larger than the grain size. Then both delta and  $d_s$  can reach some considerable fraction of the system. The same applies for the length scale *L*. We add this statement after introducing the length scale. At various relevant places we indicate that  $d_s$  is the grain size *or* channel spacing. In the discussion section 5.3 we clearly introduce  $d_s$  as grain size or channel spacing and review values from grain sizes up to channel distances of 300 m. From all these number we obtain the regimes plotted in Figure 4 and everything is consistent. Of course, the regimes extend far into the region of G< 0.01, which we now emphasize more clearly.

3.) Also, as shown in figure 4, the degree of disequilibrium is proportional to G in all cases. In Regime 1, DeltaTmax is G\*z where G is small, the maximum value of z will be large. However, in Regimes 2 and 3, when G is small, unless Pe is very large, the degree of disequilibrium is likely to be very small based on the scaling laws in these two regimes since (1-exp(-z)) and M(z) are both of order 1,

I don't fully understand the concern. If you are concerned about the fact that the proportionality to *G* given in the formulas is not equally visible in the three regimes, here some explanation: In regime 1 one does'nt see the proportionality to G because increasing G decreases the maximum value of z which is equal H = 1/G cancelling the G-proportionality. In regime 2 (1-exp(-z)) is of order 1 indeed, and one clearly sees the proportionality to G, in regime 3 M is both a function of G and z, and the proportionality to G gradually disappears. I added two sentences in the text along these explanations.

**4.) Issues marked in the text**

Thanks, changes are applied appropriately.

4.1) Time scales: I think that the most important question in application to the real earth is "How large is the degree of disequilibrium likely to be?" This has implications for whether the solid rock will melt or the liquid will freeze. Since the authors have calculated a dimensionless DeltaTmax, they should also be able to estimate a magnitude of the temperature disequilibrium.

You are right, that is the main point of the whole paper, and we did'nt give a number. In the section on application to magmatic systems we add a paragraph in which we estimate the degree of thermal non-equilibrium in Kelvin for our example of mid-ocean ridges.

**Review Wilson**

**1.) Non-dimensional Numbers**

The non-dimensional parameters controlling the system has now been reduced to Pe,  $\varphi$  and G and only Pe and G in the analytical case. However, models are still presented in terms of H. Even though this is just the reciprocal of G it would be great to use a single labeling system. G and H are also listed in table 1 as dimensional numbers but, I think, only discussed in their non-dimensional form (except maybe when discussing the scale length L). Even though primes are explicitly dropped this might be a little confusing.

Fair enough, but as we cannot abandon *H* completely, we replaced *H* by *G* where possible, but keep *H* where explicitly the height of the model is addressed. Also, the labeling in Fig. 6 has been changed in favor of *G*. In Table 1 we added "mostly non-dimensional" for *G* and *H* and put the dimensions is parantheses.

**2.) Initial & Boundary Conditions**

The dependence on G' (or H') really emphasized to me how dependent these solutions are on the initial and boundary conditions selected (I guess most problems are!). Though some discussion has been added to section 2.3 about the need to specify boundary conditions in general, this isn't really the kind of physical justification for this specific problem that I was hoping for. This model imagines an initially stagnant fluid in thermal equilibrium with a solid with a (Cartesian) conductive temperature profile. The fluid then moves and while the upper boundary allows advective outflow of heat in the fluid it maintains a fixed thermal gradient in both the solid and fluid while doing so. (All of this is much more clearly stated in the updated manuscript, thank you.) The analytical model then seeks the maximum thermal disequilibrium, which occurs in the early stages of the evolution of the model and at the top of the domain. I understand that these decisions were made to make the analytical solution possible but if there is a physical scenario in which this is likely then it would be great to discuss it in section 2.3.

The authors demonstrate numerically the effects that different initial and boundary conditions might have in section 5.1.2. This only seems to consider applying the same conditions to both the solid and fluid temperature. Have the authors considered applying different conditions to the two fields? This may make a more physical scenario possible. For example the fluid temperature could use the open boundary condition, mimicking fluid escape, while the solid could use a Robin condition relating the gradient at the top boundary to an imagined crustal conductive profile above the domain depending on the temperature at the top boundary, e.g.  $\frac{\partial T}{\partial z} = \frac{T_{surf} - T}{h_{crust}}$ . Admittedly this would introduce an extra parameter for the "crust" thickness  $h_{crust}$  and the surface temperature  $T_{surf}$ .

Good point, thanks. We did'nt think of applying different boundary conditions to the fluid and solid. We now include this possibility in section 2.3 as a possibility, particularly the suggested open condition for the fluid and the Robin lid condition for the solid. Yet I am not sure whether such a Robin condition is physically better in view of relatively rapid temperature variations: It is unphysical to instantaneously change the slope within the complete lid. The temperature in the imagined lid can only vary on the diffusive timescale of the lid, which is much longer than all time scales in our model if the lid is thicker than *H*. We have done some models with the proposed boundary conditions (open for the fluid, Robin lid

for the solid) and include a paragraph describing that for thick lids (> *H* as supposed for our scenarios) the temperatures evolve generally similar to our flux condition. And we explain the differences. In Fig. 2 here we show this series of models, but don't want to expand the paper even more by including another figure with a fifth boundary condition.

Fig. 2. Models with an open boundary condition at the top for the fluid and a Robin boundary condition for the solid representing an imagined lid of thickness  $H_{lid}$ . The columns show models with same parameters as those of Fig. 6 in the paper. The rows contain models with different lid thicknesses.

**3.) Time Evolution & Steady State**

I appreciated the authors' efforts to clarify the time evolution in the figures by giving some time increments and distinguishing the lines a little. In many cases the total time was listed for all models. Could the authors confirm that in the time evolution figures (e.g. Figure 6) the steady state solution (as opposed to just the solution at an arbitrary final time) is included in all subplots?

We confirm that all models with G = 0.1 have been run close to steady state, which is reached at about t = 100, while the cases with G = 0.01 have not. They would need to be run until t = 10000, i.e. 10 to 20 times longer than we did. We don't think it is necessary to rerun those models because we don't learn too much about thermal non-equilibrium which is reached much earlier during stage 1. We include statements about reaching steady state or not where appropriate.

4.) Even More Minor Points

There are still quite a few small typos throughout the paper that should be corrected in the editing phase. Below are a few things that caught my eye:

- line 31: "On small scale" → "On a small scale"
- line 67: "or more general"  $\rightarrow$  "or more generally"
- *line 113: Perhaps introduce a new paragraph after "Primed quantities are non-dimensional."?*
- *line 179: is the second statement about resolution only necessary for a few of the numerical cases presented in figure 3?*
- equations 21, 22, 23: G appears inconsistently italicized in these three equations (see line 331 also)
- equations 32 & 33: the non-dimensionalized prime is placed inconsistently throughout the variables in these equations, sometimes as a subscript and also as a superscript
- table 1, last line: "reference. respectively" → "reference, respectively"?
- figure 3: could figure S1 be incorporated into figure 3a to avoid having to reference a supplemental figure? Also, while figure 3 is necessary to show the numerical results could it reference figure 4 (and vice versa) as it (I believe) represents slices through the latter?

Thanks, we have included all corrections and suggestions. As for eq. 32 & 33, they are all superscripts. Hope this will be correctly displayed after editing.

**Another point not noted by the reviewers but by the authors**

In section 5.1.3 the formulation addresses different physical properties of the two phases. In the previous version we stated that this formulation adds three new non-dimensional numbers. In fact, modifying the scaling by combining the non-dimensional effective thermal conductivity with the length scale introduces a new alternative Peclet number and reduces the number of new non-dimensional numbers to two rather than three. Thus, equations (32) and (33) are now rewritten in terms of this modified scaling. The shown models and discussion are not affected as we assume  $\lambda_{eff}' = 1$  so that the new and old Peclet numbers are identical.